# Physics-Informed Weakly Supervised Learning for Interatomic Potentials

## Abstract

Machine learning plays an increasingly important role in computational chemistry and materials science, complementing computationally intensive ab initio and first-principles methods. Despite their utility, machine-learning models often lack generalization capability and robustness during atomistic simulations, yielding unphysical energy and force predictions that hinder their real-world applications. We address this challenge by introducing a physics-informed, weakly supervised approach for training machine-learned interatomic potentials (MLIPs). We introduce two novel loss functions, extrapolating the potential energy via a Taylor expansion and using the concept of conservative forces. Our approach improves the accuracy of MLIPs applied to training tasks with sparse training data sets and reduces the need for pre-training computationally demanding models with large data sets. Particularly, we perform extensive experiments demonstrating reduced energy and force errors—often lower by a factor of two—for various baseline models and benchmark data sets. Moreover, we demonstrate improved robustness during MD simulations of the MLIP models trained with the proposed weakly supervised loss. Finally, we show that our approach facilitates MLIPs' training in a setting where the computation of forces is infeasible at the reference level, such as those employing complete-basis-set extrapolation. An implementation of our method and scripts for executing experiments are available at `https://anonymous.4open.science/r/PICPS-ML4Sci-1E8F`.

## 1 Introduction

Ab initio and first-principles methods are inevitable for the computer-aided exploration of molecular and material properties used in the chemical sciences and engineering (Parrinello, 1997; Carloni et al., 2002; Iftimie et al., 2005). However, commonly employed ab initio and first-principles approaches—such as coupled cluster (CC) (Purvis & Bartlett, 1982; Bartlett & Musiał, 2007) and density functional theory (DFT) (Hohenberg & Kohn, 1964; Kohn & Sham, 1965), respectively—require substantial compute resources. Thus, they typically allow only for atomistic simulations of small- to medium-sized atomic systems and restrict the accessible simulation times, which affects the accuracy of estimated molecular and material properties. Classical force fields can extend these length and time scales, providing a computationally efficient alternative to first-principles approaches, but often lack accuracy. Machine-learning-based models hold promise to bridge the gap between first-principles and classical approaches, yielding computationally efficient and accurate machine-learned interatomic potentials (MLIPs) (Smith et al., 2017; Chanussot et al., 2021; Unke et al., 2021; Merchant et al., 2023; Kovács et al., 2023; Batatia et al., 2023).

These MLIPs, however, face several challenges. They require the generation of training data sets that sufficiently cover configurational (atom positions) and compositional (atom types) spaces using, e.g., molecular dynamics (MD) simulations based on ab initio or first-principles approaches. Given the high computational cost of the commonly used data generation approaches, the resulting training data sets are often sparse and prevent the application of MLIPs to new molecular and material systems. Active learning can be used to address this challenge (Li et al., 2015; Vandermause et al., 2020; Zaverkin et al., 2024b), but still requires generating a non-negligible number of first-principles (i.e., DFT) data to train the initial model, which is then used to explore the phase space with sufficiently long MD simulations. Hence, a strong motivation for the proposed method is its use in combination with active learning to acquire additional training data. Furthermore, MLIPs often lack sufficient

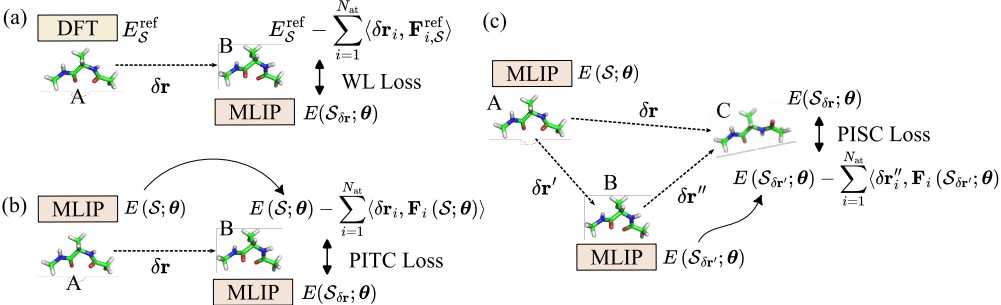

Figure 1: **Schematic illustration of physics-informed weakly supervised losses used in this work.** (a) Taylor-expansion-based weak label (WL) loss with approximate labels obtained from reference energies and atomic forces (Cooper et al., 2020). (b) Physics-inspired Taylor-expansion-consistency (PITC) loss with approximate labels obtained from energies and atomic forces predicted by an MLIP. (c) Physics-inspired spatial consistency (PISC) loss with approximate labels obtained from energies and atomic forces predicted by an MLIP. Here, $E(\mathcal{S}; \boldsymbol{\theta})$ and $\mathbf{F}_i(\mathcal{S}; \boldsymbol{\theta})$ denote the potential energy and atomic forces predicted by an MLIP parametrized by $\boldsymbol{\theta}$, $\mathcal{S}$ and $\mathcal{S}_{\delta\mathbf{r}}$ define the original atomic structure and the one perturbed by $\delta\mathbf{r}$.

generalization capability and robustness during MD simulations, i.e., they are sensitive to outliers and local perturbations of atomic structures. This sensitivity of ML models is caused by existing data sets and data generation techniques not providing sufficient coverage of configurational and compositional spaces.

**Contributions.** This paper addresses these challenges using a physics-informed weakly supervised learning (PIWSL) approach. Our method is designed to learn an MLIP, which can accurately predict the potential energy and atomistic forces for an atomic system exposed to local perturbations. In particular, our contributions are as follows: (i) We introduce PIWSL based on basic physical principles, such as the concept of conservative forces. We combine it with extrapolating the potential energy via a Taylor expansion and derive two novel physics-informed loss functions, schematically illustrated in figure 1 (b) and (c). Particularly, we obtain physics-informed Taylor-expansion-based consistency (PITC) and physics-informed spatial consistency (PISC) losses, which build the basis for the PIWSL approach. (ii) By conducting extensive experiments, we demonstrate that PIWSL facilitates the training of MLIPs without access to large training data sets. Furthermore, we show that MLIPs trained with PIWSL are more robust during MD simulations compared to those trained without it. (iii) We also observe that PIWSL improves accuracy in predicted total energies and atomic forces, even without access to force labels. This scenario is expected when training MLIPs with reference methods for which calculating atomic forces is infeasible (Smith et al., 2019; 2020; Zaverkin et al., 2023). Thus, our results open new possibilities for training MLIPs using highly accurate energy labels, such as those obtained by extrapolating CCSD(T) energies to the complete basis set (CBS) limit (Hobza & Šponer, 2002; Feller et al., 2006). (iv) Finally, PIWSL mitigates sensitivity issues associated with limited sizes of available data sets by taking into account the potential energy response to local perturbations in atomic structures.

## 2 RELATED WORK

**Machine-Learned Interatomic Potentials.** There is a growing interest in using ML-based models for investigating molecular and material systems as they allow performing atomistic simulations with an accuracy on par with first-principles methods but at a fraction of the computational cost. The field of machine-learned interatomic potentials (MLIPs) emerged over two decades ago (Blank et al., 1995) and has been one of the most active research directions since then (Behler & Parrinello, 2007; Artrith et al., 2011; Artrith & Urban, 2016; Smith et al., 2017; Shapeev, 2016; Schütt et al., 2017; Thomas et al., 2018; Unke & Meuwly, 2019; Drautz, 2019; Zaverkin & Kästner, 2020; Zaverkin et al., 2021; Thomas et al., 2018; Schütt et al., 2021; Shuaibi et al., 2021a; Passaro & Zitnick, 2023; Liao et al., 2023; Batzner et al., 2022; Musaelian et al., 2023; Batatia et al., 2022; Zaverkin et al., 2024a). The development of local higher-body-order representations (Shapeev, 2016; Drautz, 2019; Zaverkin

& Kästner, 2020; Zaverkin et al., 2021) and the emergence of equivariant message-passing neural networks (MPNNs) (Thomas et al., 2018; Schütt et al., 2021; Shuaibi et al., 2021a; Passaro & Zitnick, 2023; Liao et al., 2023; Batzner et al., 2022; Musaelian et al., 2023; Batatia et al., 2022; Zaverkin et al., 2024a) significantly advanced the field. These methods enable the cost-efficient generation of accurate MLIPs for modeling interactions in many-body atomic systems and account for crucial inductive biases as the invariance of the potential energy under rotation.

**Physics-Informed Machine Learning.** Physics-informed ML aims to model physical systems using data-driven techniques and incorporates physics principles into ML-based models. For example, MLIPs based on equivariant MPNNs enforce the invariance of the potential energy under rotation and use equivariant features to enrich the building of many-body contributions to it (Thomas et al., 2018; Batzner et al., 2022; Batatia et al., 2022; Musaelian et al., 2023; Liao et al., 2023). Furthermore, physics constraints can be integrated via auxiliary loss functions, prompting ML models to learn important physical relationships, as demonstrated for physics-informed neural networks (PINNs) (Raissi et al., 2019; Cai et al., 2022), which learn to model solutions of partial differential equations by minimizing residuals during training. Applying physics-informed ML to molecular modeling has gained attraction in both ML and computational chemistry communities (Godwin et al., 2022; Ni et al., 2024). As such, prior work (Cooper et al., 2020) has motivated our current research and is discussed in more detail in subsequent sections.

# 3 BACKGROUND AND PROBLEM DEFINITION

**Machine-Learned Interatomic Potentials.** An atomic configuration, denoted as $\mathcal{S} = \{\mathbf{r}_i, Z_i\}_{i=1}^{N_{\text{at}}}$, contains $N_{\text{at}}$ atoms and is defined by atom positions $\mathbf{r}_i \in \mathbb{R}^3$ and atom types $Z_i \in \mathbb{N}$. We consider mapping atomic configurations to scalar energies, i.e., $f_{\boldsymbol{\theta}} : \mathcal{S} \mapsto E \in \mathbb{R}$ with $\boldsymbol{\theta}$ denoting trainable parameters. We define $E(\mathcal{S}; \boldsymbol{\theta})$ as the energy predicted by an MLIP for an atomic configuration $\mathcal{S}$. For most MLIPs, atomic forces are computed as the negative gradients of the potential energy with respect to atom positions, i.e., $\mathbf{F}_i(\mathcal{S}; \boldsymbol{\theta}) = -\nabla_{\mathbf{r}_i} E(\mathcal{S}; \boldsymbol{\theta})$. In this way, these MLIPs ensure that the resulting forces are conservative (curl-free) and the total energy is conserved during a dynamic simulation. However, some models are designed to predict atomic forces directly (Hu et al., 2021; Passaro & Zitnick, 2023; Liao et al., 2023; Chanussot et al., 2021). While this approach avoids expensive gradient computations, it violates the law of energy conservation (Chmiela et al., 2017).

Trainable parameters $\boldsymbol{\theta}$ are optimized by minimizing loss functions on training data $\mathcal{D}$ comprising a total of $N_{\text{train}}$ atomic configurations $\{\mathcal{S}^{(k)}\}_{k=1}^{N_{\text{train}}}$ as well as their energies $\{E_{\mathcal{S}}^{\text{ref}}\}_{\mathcal{S} \in \mathcal{D}}$ and atomic forces $\{\{\mathbf{F}_{i,\mathcal{S}}^{\text{ref}}\}_{i=1}^{N_{\text{at}}}\}_{\mathcal{S} \in \mathcal{D}}$

$$\mathcal{L}(\mathcal{D}; \boldsymbol{\theta}) = \sum_{\mathcal{S} \in \mathcal{D}} L(S; \boldsymbol{\theta}) = \sum_{\mathcal{S} \in \mathcal{D}} \left[ C_{\text{e}} \ell\left(E(\mathcal{S}; \boldsymbol{\theta}), E_{\mathcal{S}}^{\text{ref}}\right) + C_{\text{f}} \sum_{i=1}^{N_{\text{at}}} \ell\left(\mathbf{F}_i(\mathcal{S}; \boldsymbol{\theta}), \mathbf{F}_{i,\mathcal{S}}^{\text{ref}}\right) \right]. \quad (1)$$

Here, $\ell$ denotes a point-wise loss function such as the absolute and squared error between the predicted and reference total energies and atomic forces. Typically, reference energies $E_{\mathcal{S}}^{\text{ref}}$ and atomic forces $\mathbf{F}_{i,\mathcal{S}}^{\text{ref}}$ are provided by ab initio or first-principles methods such as CC or DFT, respectively. The relative contributions of energies and forces in Eq. (1) are balanced with the coefficients $C_{\text{e}}$ and $C_{\text{f}}$.

**Weakly Supervised Learning.** Generating many reference labels with a first-principles approach is challenging due to the high computational cost. Furthermore, the calculation of atomic forces can be infeasible for some high-accuracy ab initio methods, e.g., for CCSD(T)/CBS. In this work, we focus on weakly supervised learning methods to improve the performance of MLIPs in scenarios when only a limited amount of data is available. These involve the generation of approximate but physically motivated total energies for atomic structures generated by small perturbations of their atomic positions, i.e., $\mathcal{S}_{\delta \mathbf{r}} = \{\mathbf{r}_i + \delta \mathbf{r}_i, Z_i\}_{i=1}^{N_{\text{at}}}$ with a perturbation vector $\delta \mathbf{r}$, where $\delta \mathbf{r}_i$ is the perturbation vector for atom $i$. Approximate labels are computed with MLIPs during their training.

# 4 PHYSICS-INFORMED WEAKLY SUPERVISED LEARNING

For MLIPs, the generation of approximate labels employed in weakly supervised losses is highly non-trivial. Small perturbationss in atomic structures can lead to significant changes in energies and

atomic forces. Thus, standard approaches that are effective for many ML tasks (Yang et al., 2022) are typically not applicable to MLIPs. To address this problem, we propose a physics-informed weakly supervised learning approach that involves (i) a Taylor expansion of the potential energy for computing the response to atomic perturbations and (ii) spatial consistency to estimate the displaced potential energy based on the concept of conservative forces. We finally introduce the PIWSL loss term, combining both classes of weakly supervised loss functions with the supervised loss.

### 4.1 PHYSICS-INFORMED TAYLOR-EXPANSION-BASED CONSISTENCY LOSS

This section introduces the physics-informed Taylor-expansion-based consistency (PITC) loss. Particularly, we relate the energy predicted directly for a displaced atomic configuration with the energy obtained by the Taylor expansion from the original configuration; see figure 1 (b). We estimate the energy for an atomic structure $\mathcal{S}$ drawn from the training data set with atomic positions displaced by a vector $\delta \mathbf{r}$: $\mathcal{S}_{\delta \mathbf{r}} = \{\mathbf{r}_i + \delta \mathbf{r}_i, Z_i\}_{i=1}^{N_{\mathrm{at}}}$. For this atomic configuration, we expand the energy predicted by an MLIP in its first-order Taylor series around the atomic perturbation vector $\delta \mathbf{r}_i$ and obtain

$$E\left(\mathcal{S}_{\delta \mathbf{r}}; \boldsymbol{\theta}\right) \approx E\left(\mathcal{S}; \boldsymbol{\theta}\right) - \sum_{i=1}^{N_{\mathrm{at}}} \langle \delta \mathbf{r}_i, \mathbf{F}_i\left(\mathcal{S}; \boldsymbol{\theta}\right) \rangle + \mathcal{O}\left(\|\delta \mathbf{r}\|^2\right), \tag{2}$$

where $\langle \cdot \rangle$ denotes the inner product. Here, we used that atomic forces are defined as the negative gradients of the potential energy. For small magnitudes of $\delta \mathbf{r}_i$, the second order term $\mathcal{O}\left(\|\delta \mathbf{r}\|^2\right)$ in Eq. (2) can be neglected. Using approximate labels $E\left(\mathcal{S}_{\delta \mathbf{r}}; \boldsymbol{\theta}\right)$, we define the PITC loss as

$$L_{\mathrm{PITC}}\left(\mathcal{S}; \boldsymbol{\theta}\right) = \ell\left(E\left(\mathcal{S}_{\delta \mathbf{r}}; \boldsymbol{\theta}\right), E\left(\mathcal{S}; \boldsymbol{\theta}\right) - \sum_{i=1}^{N_{\mathrm{at}}} \langle \delta \mathbf{r}_i, \mathbf{F}_i\left(\mathcal{S}; \boldsymbol{\theta}\right) \rangle\right), \tag{3}$$

where $\ell$ denotes a point-wise loss for regression problems and $\delta \boldsymbol{r}$ is randomly-sampled or determined adversarially; see section 4.4 for more details. Hence, whenever we encounter a structure $\mathcal{S}$ in a batch during training, a new $\delta \boldsymbol{r}$ is computed for each $\mathcal{S}$.

### 4.2 PHYSICS-INFORMED SPATIAL-CONSISTENCY LOSS

This section introduces a physics-informed approach for generating weak labels based on the concept of conservative forces. Thus, we leverage that the energy difference between two points on the potential energy surface is independent of the path taken between them. We consider two paths from a reference point to the same target point, composed of three perturbation vectors in total. We estimate the potential energy at the target point via Eq. (2). An example of two paths is demonstrated in figure 1 (c). The figure relates the energy obtained when displacing atomic positions of the original configuration $\mathcal{S}$ (denoted by A in the figure) by $\delta \mathbf{r}$ (from configuration A to C) with the energy obtained through consecutive perturbations $\delta \mathbf{r}'$ (from configuration A to B) and $\delta \mathbf{r}''$ (from configuration B to C).

For the first path, we directly predict the energy with an MLIP, i.e., $E\left(\mathcal{S}_{\delta \mathbf{r}}; \boldsymbol{\theta}\right)$, which is related to the approximated energy at $\mathbf{r} + \delta \mathbf{r}$ using Eq. (3) through PITC loss. For the second path, we directly compute the energy $E\left(\mathcal{S}_{\delta \mathbf{r}'}; \boldsymbol{\theta}\right)$ for atomic positions displaced by $\delta \mathbf{r}'$ and use it to approximate $E\left(\mathcal{S}_{\delta \mathbf{r}}; \boldsymbol{\theta}\right)$ after applying the second perturbation vector $\delta \mathbf{r}'' \equiv \delta \mathbf{r} - \delta \mathbf{r}'$. The physics-informed spatial consistency (PISC) loss can be defined as

$$L_{\mathrm{PISC}}\left(\mathcal{S}; \boldsymbol{\theta}\right) = \ell\left(E\left(\mathcal{S}_{\delta \mathbf{r}}; \boldsymbol{\theta}\right), E\left(\mathcal{S}_{\delta \mathbf{r}'}; \boldsymbol{\theta}\right) - \sum_{i=1}^{N_{\mathrm{at}}} \langle \delta \mathbf{r}_i'', \mathbf{F}_i\left(\mathcal{S}_{\delta \mathbf{r}'}; \boldsymbol{\theta}\right) \rangle\right), \tag{4}$$

where $\delta \boldsymbol{r}$ is randomly-sampled or determined adversarially; see section 4.4. After joint training of PITC and PISC losses, the three different estimations at $\mathcal{S}_{\delta \mathbf{r}}$ become spatially consistent. Note that our conservative forces-based approach is not limited to relations between two perturbation paths or three perturbation vectors. We discuss several other possible configurations in section F.3.

### 4.3 COMBINED PHYSICS-INFORMED WEAKLY SUPERVISED LOSS

Together with the usual MLIP loss function given in Eq. (1), the overall objective, which we refer to as the PIWSL loss, can be written as

$$\arg\min_{\boldsymbol{\theta}} \tilde{\mathcal{L}}(\mathcal{D}; \boldsymbol{\theta}) = \arg\min_{\boldsymbol{\theta}} \sum_{\mathcal{S} \in \mathcal{D}} \left( L(\mathcal{S}; \boldsymbol{\theta}) + C_{\text{PITC}} L_{\text{PITC}}(\mathcal{S}; \boldsymbol{\theta}) + C_{\text{PISC}} L_{\text{PISC}}(\mathcal{S}; \boldsymbol{\theta}) \right), \quad (5)$$

where $C_{\text{PITC}}$ and $C_{\text{PISC}}$ are the weights of the weakly supervised PITC and PISC losses.

### 4.4 DETERMINING PERTURBATION DIRECTIONS AND MAGNITUDES

The effectiveness of the proposed approach depends on appropriate choices of the perturbation vectors $\delta\mathbf{r}$. We introduce and justify various strategies for generating the perturbations used in Eq. (3) and Eq. (4). Any vector $\delta\mathbf{r}$ can be written as $\delta\mathbf{r} \equiv \epsilon\mathbf{g}/\|\mathbf{g}\|_2$, where $\epsilon$ is the magnitude of $\delta\mathbf{r}$ and $\mathbf{g}/\|\mathbf{g}\|_2$ is the direction of $\delta\mathbf{r}$. Physical constraints can limit $\epsilon$. Specifically, we can obtain the maximum perturbation length from the validity of the Taylor expansion in Eq. (2), which, as discussed in section 5.3, is typically given as at most 30% of the original bond length whose shortest example is the bond between carbon and hydrogen atoms, about 1.09 Å; see also figure 2 (c) and (d). The specific values of $\epsilon$ chosen for our experiments are provided in section D.1.

To determine $\mathbf{g}/\|\mathbf{g}\|_2$ we explore two strategies. First, we compute it as the unit vector of a perturbation vector sampled from the uniform distribution on the interval $(-1, 1)$ for each direction

$$\delta\mathbf{r}_{\text{rnd}} \equiv \epsilon\mathbf{g}/\|\mathbf{g}\|_2. \quad (6)$$

Second, we compute an adversarial direction, as proposed by Goodfellow et al. (2014); Miyato et al. (2018), which involves defining it as the direction (the gradients) in which the loss error increases the most at the current atom coordinates $\mathbf{r}$ and for the current predicted energy. Assuming the norm of adversarial perturbation as $L_2$, the adversarial direction can be approximated by (Miyato et al., 2018)

$$\delta\mathbf{r}_{\text{adv}} \equiv \epsilon\mathbf{g}/\|\mathbf{g}\|_2, \text{ where } \mathbf{g} = \nabla_{\mathbf{r}} L_{\text{dist}}(\mathbf{y}^{\text{pred}}, \mathbf{y}^{\text{ref}}), \quad (7)$$

where $L_{\text{dist}}$ is a distance measure function to be maximized by adding $\delta\mathbf{r}_{\text{adv}}$, with $\mathbf{y}^{\text{pred}}$ and $\mathbf{y}^{\text{ref}}$ being the ML model prediction and the reference values. Due to their computational efficiency, we mainly use Eq. (6) in our experiments. A quantitative comparison between the random and adversarial directions is provided in section 5.5.

## 5 EXPERIMENTS

We evaluate our method through extensive experiments with the following objectives: (1) comparing PIWSL with existing baselines, (2) analyzing the impact of the PIWSL using the aspirin molecule, including MD simulations, (3) assessing PIWSL's ability to improve energy and force predictions when force labels are inaccessible, (4) comparing with a prior weakly supervised approach, (5) an ablation study, and (6) comparing random and adversarial generation of the perturbation vector. We focus on the data-scarce setting where the number of training samples is between 100 and 1000, as generating large datasets using ab initio and first-principles approaches is computationally expensive.

### 5.1 MODELS AND DATA SETS

We trained the following representative models that are provided in the Open Catalyst code base (Chanussot et al., 2021): SchNet (Schütt et al., 2017), PaiNN (Schütt et al., 2021), SpinConv (Shuaibi et al., 2021a), eSCN (Passaro & Zitnick, 2023), and Equiformer v2 (Liao et al., 2023), covering MLIPs with a smaller (SchNet, SpinConv, PaiNN) and larger number of parameters (eSCN, Equiformer v2). Moreover, we also considered the MACE model (Batatia et al., 2022), a popular state-of-the-art model that we use to evaluate the impact of PIWSL on the MD22 data set. Unless otherwise mentioned and except for SchNet, forces are directly predicted and not computed through the negative gradient of the energy. The results where forces are computed as negative energy gradients are analyzed in section 5.4 and section F.9. To evaluate the effect and dependency of the physics-informed weakly supervised approach in detail, we performed the training on various data sets: ANI-1x as a heterogeneous molecular data set (Smith et al., 2020), TiO$_2$ as a data set for inorganic materials (Artrith & Urban,

Table 1: **Energy (E) and force (F) root-mean-square errors (RMSEs) for the ANI-1x data set.** The results are obtained by averaging over three independent runs. Energy RMSE is given in kcal/mol, while force RMSE is in kcal/mol/Å.

| | | $N_{\text{train}} = 100$ | | | $N_{\text{train}} = 1000$ | | |
| | | Baseline | Noisy Nodes | PIWSL | Baseline | Noisy Nodes | PIWSL |
|---|---|---|---|---|---|---|---|
| SchNet | E | $65.09 \pm 2.42$ | $\mathbf{57.39 \pm 0.05}$ | $60.30 \pm 1.77$ | $31.49 \pm 0.01$ | $\mathbf{31.10 \pm 0.00}$ | $31.50 \pm 0.00$ |
| | F | $29.06 \pm 0.19$ | $\mathbf{25.62 \pm 0.01}$ | $28.20 \pm 0.60$ | $18.94 \pm 0.01$ | $\mathbf{18.10 \pm 0.00}$ | $18.93 \pm 0.00$ |
| PaiNN | E | $168.01 \pm 1.22$ | $464.55 \pm 6.91$ | $\mathbf{109.89 \pm 11.46}$ | $56.62 \pm 2.80$ | $305.76 \pm 33.93$ | $\mathbf{24.53 \pm 0.48}$ |
| | F | $21.33 \pm 0.10$ | $20.82 \pm 0.03$ | $\mathbf{18.76 \pm 0.30}$ | $12.96 \pm 0.06$ | $14.25 \pm 0.18$ | $\mathbf{11.43 \pm 0.05}$ |
| SpinConv | E | $162.14 \pm 7.55$ | $147.73 \pm 2.23$ | $\mathbf{130.97 \pm 8.58}$ | $43.59 \pm 1.71$ | $299.33 \pm 419.10$ | $\mathbf{39.44 \pm 1.31}$ |
| | F | $21.22 \pm 0.43$ | $\mathbf{21.08 \pm 0.43}$ | $21.61 \pm 0.44$ | $14.51 \pm 1.07$ | $15.83 \pm 0.75$ | $\mathbf{13.59 \pm 0.20}$ |
| eSCN | E | $214.52 \pm 7.55$ | $521.92 \pm 12.05$ | $\mathbf{183.70 \pm 9.79}$ | $59.59 \pm 8.92$ | $241.34 \pm 20.16$ | $\mathbf{21.03 \pm 0.56}$ |
| | F | $20.07 \pm 0.27$ | $23.68 \pm 0.11$ | $\mathbf{19.69 \pm 0.05}$ | $12.50 \pm 0.78$ | $14.42 \pm 0.84$ | $\mathbf{11.83 \pm 0.12}$ |
| Equiformer | E | $398.71 \pm 13.69$ | $632.38 \pm 0.11$ | $\mathbf{154.98 \pm 8.83}$ | $54.52 \pm 4.52$ | $854.33 \pm 317.7$ | $\mathbf{20.89 \pm 0.50}$ |
| | F | $20.71 \pm 0.05$ | $21.82 \pm 0.01$ | $\mathbf{20.55 \pm 0.05}$ | $10.10 \pm 0.00$ | $24.79 \pm 2.05$ | $\mathbf{9.68 \pm 0.03}$ |

2016), the revised MD17 (rMD17) data set containing small molecules with sampled configurational spaces for each (Chmiela et al., 2017; 2018; Christensen & von Lilienfeld, 2020), the MD22 data set containing large molecules Chmiela et al. (2023), and LMNTO as another material data set (Cooper et al., 2020); the results for rMD17, MD22, and LMNTO are provided in section F.1. The detailed description of each data set is provided in section D.3.

## 5.2 BENCHMARK RESULTS

We compare models trained using the PIWSL loss (see Eq. (5)) with baseline models trained using the standard supervised loss only (see Eq. (1)). We also compare our approach to a recently proposed data augmentation method that incorporates the task of denoising random perturbations of the atomic coordinates into the learning objective (NoisyNode) (Godwin et al., 2022). More details on the setup are provided in section D.1. In the following, all evaluation metrics are computed for the test data set.

**Heterogeneous Molecular Data Set (ANI-1x).** The results provided in Table 1 show that our approach improves the baseline models' performance in almost all cases. In particular, the error reduction for the predicted energies is often between 10 % and more than 50 %. Interestingly, we observe an improved accuracy for potential energies and atomic forces because we include force prediction in PITC and PISC losses, different from the previous work (Cooper et al., 2020). In most cases, except for SchNet, the data augmentation method (NoisyNode) deteriorates the accuracy of the MLIPs because it does not incorporate the proper response of the energy and atomic forces to the perturbation of atomic positions.

**Training Data Set Size Dependence (ANI-1x).** We train MLIPs with training set sizes of $[50, 10^2, 10^3, 10^4, 10^5, 10^6, 5 \times 10^6]$[1]. The results are plotted in figure 2 (a) and (b). Although the observed error reduction depends strongly on the type of MLIP used, the benefit of the weakly supervised losses often decreases slightly with the number of training samples. This result can be expected as the area covered by the weakly supervised losses is also gradually covered by the reference data as the number of training samples increases. Moreover, the gain in accuracy of energy predictions is more significant than that for forces trained only indirectly through the consistency constraint in PITC; see Eq. (3). Finally, it is shown that the improvement is more significant for highly parameterized MLIPs, which benefit the most from increasing the training data size through PIWSL.

**Inorganic Bulk Materials (TiO$_2$).** Titanium dioxide (TiO$_2$) is a highly relevant metal oxide for industrial applications, featuring several high-pressure phases. Thus, ML models should be able to predict total energies and atomic forces for various high-pressure phases of TiO$_2$, considering periodic boundaries (relevant when aggregating over the local atomic neighborhood). The results for trained models are provided in Table 2. Similar to the ANI-1x data set, our approach improves the accuracy of predicted energies and atomic forces. Interestingly, although the error in the potential

---

[1]The results for training data sizes of $10^5$, $10^6$, and $5 \times 10^6$ are provided in section F.1.

Table 2: **Energy (F) and force (F) root-mean-square errors (RMSEs) for the TiO$_2$ data set.** The results are obtained by averaging over three independent runs. Energy RMSE is given in kcal/mol, while force RMSE is in kcal/mol/Å.

| | | | $N_{\text{train}} = 100$ | | | $N_{\text{train}} = 1000$ | |
| | | Baseline | Noisy Nodes | PIWSL | Baseline | Noisy Nodes | PIWSL |
|---|---|---|---|---|---|---|---|
| SchNet[a] | E | $17.21 \pm 0.00$ | $19.68 \pm 0.00$ | $\mathbf{17.08 \pm 0.00}$ | $9.56 \pm 0.00$ | $9.60 \pm 0.00$ | $\mathbf{9.51 \pm 0.00}$ |
| | F | $2.84 \pm 0.00$ | $\mathbf{2.70 \pm 0.00}$ | $2.83 \pm 0.00$ | $2.14 \pm 0.00$ | $2.15 \pm 0.00$ | $\mathbf{2.13 \pm 0.00}$ |
| PaiNN[b] | E | $14.41 \pm 0.16$ | n/a[b] | $\mathbf{13.95 \pm 0.09}$ | $4.49 \pm 0.15$ | n/a[b] | $\mathbf{3.63 \pm 0.20}$ |
| | F | $1.59 \pm 0.01$ | n/a[b] | $\mathbf{1.56 \pm 0.01}$ | $0.41 \pm 0.02$ | n/a[b] | $\mathbf{0.34 \pm 0.01}$ |
| SpinConv | E | $20.00 \pm 0.42$ | $18.76 \pm 0.74$ | $\mathbf{16.98 \pm 0.99}$ | $4.17 \pm 0.76$ | $4.09 \pm 0.65$ | $\mathbf{2.50 \pm 0.40}$ |
| | F | $1.58 \pm 0.03$ | $\mathbf{1.53 \pm 0.03}$ | $1.59 \pm 0.03$ | $0.65 \pm 0.02$ | $0.71 \pm 0.16$ | $\mathbf{0.58 \pm 0.05}$ |
| eSCN | E | $16.41 \pm 1.10$ | $20.92 \pm 0.00$ | $\mathbf{12.63 \pm 0.78}$ | $3.31 \pm 1.18$ | $20.90 \pm 0.01$ | $\mathbf{1.40 \pm 0.10}$ |
| | F | $1.57 \pm 0.04$ | $1.66 \pm 0.00$ | $\mathbf{1.44 \pm 0.03}$ | $0.46 \pm 0.23$ | $1.66 \pm 0.00$ | $\mathbf{0.21 \pm 0.00}$ |
| Equiformer | E | $18.21 \pm 0.02$ | $19.06 \pm 0.02$ | $\mathbf{13.93 \pm 0.09}$ | $3.67 \pm 0.03$ | $18.75 \pm 0.05$ | $\mathbf{1.82 \pm 0.34}$ |
| | F | $1.56 \pm 0.01$ | $1.64 \pm 0.00$ | $\mathbf{1.51 \pm 0.19}$ | $\mathbf{0.17 \pm 0.01}$ | $1.58 \pm 0.00$ | $\mathbf{0.17 \pm 0.01}$ |

[a] We used a larger batch size of 32 for SchNet since we obtained extremely high errors for the batch size of 4. A more detailed discussion of the experimental results for SchNet is provided in section F.1.
[b] Because of a numerical instability of PaiNN when perturbing atomic coordinates, the cutoff radius is reduced from 12 Å to 5 Å in this experiment. Predicted values become n/a when atomic configurations are perturbed.

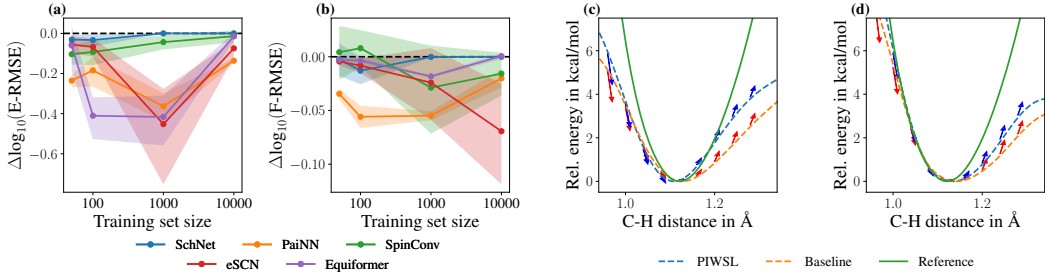

Figure 2: **(a, b) Relative performance gains for MLIPs trained with PIWSL compared to those trained without it and (c, d) Potential energy profiles for a C–H bond of the aspirin molecule.** Relative performance gains are evaluated for (a) energy (E-) and (b) force (F-) RMSEs. These results are presented for the ANI-1x data set. Potential energy profiles for a C–H bond of the aspirin molecule are presented for models trained using (c) 100 and (d) 200 configurations. The red and blue arrows indicate the direction from the original structure ($E(\mathcal{S}; \boldsymbol{\theta})$) to the perturbed one ($E(\mathcal{S}_{\delta\mathbf{r}}; \boldsymbol{\theta})$), as defined by Eq. (2), for the baseline and PIWSL model predictions, respectively.

energy for 1000 training configurations reaches small RMSE values, from 2 to 4 kcal/mol in predicted energy, the PIWSL still provides a further error reduction. This observation indicates strong evidence of the effectiveness of PIWSL applied to bulk materials.

## 5.3 QUALITATIVE IMPACT OF PIWSL

We evaluate the prediction variance and robustness of an MLIP model trained with PIWSL using the aspirin molecule, focusing on the potential energy's dependence on the C–H bond length. In this work, robustness refers to the prediction robustness of an MLIP to perturbations in atomic coordinates. In the literature, the robustness of MLIPs also means their stability during MD simulations.

We train PaiNN on the rMD17 aspirin data set using 100 and 200 configurations with and without the PIWSL loss. The detailed training setup and errors of the used MLIPs are summarized in section F.4.1. We examine the potential energy varying the length of a C–H bond from 0.9 Å to 1.4 Å. The equilibrium C–H bond length is about 1.09 Å. The results in figure 2 (c) and (d) demonstrate that

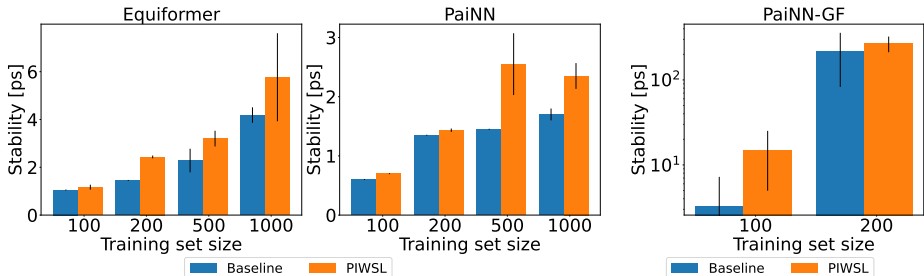

Figure 3: **Stability analysis of the MLIP models during MD simulations.** Stability during MD simulations is assessed for the baseline MLIP models and those trained with PIWSL. Left: The direct force prediction model. Right: The gradient-based force prediction model. All results are obtained for the aspirin molecule and MD simulations in the microcanonical (NVE) statistical ensemble. We measure stability during MD simulations according to Fu et al. (2023).

the PIWSL method improves the predicted potential energy profile, indicating improved robustness under perturbations of atom coordinates.

Although the estimated potential energies do not always match the reference values, the direction between the original and perturbed configurations, indicated by arrows in figure 2, consistently follows the gradient of the reference potential energy, corresponding to the negative force. In figure 2, we use a perturbation length of $||\delta \mathbf{r}|| = 0.01$ Å. This consistency with the energy gradient underscores the PIWSL method's effectiveness, ensuring the alignment between predicted total energies and force and improving the corresponding RMSE values. As discussed in section 4.2, PIWSL also addresses the limitation of MLIPs that employ separate force branches and do not guarantee the prediction of conservative forces. The proposed method reduces the curl of predicted forces, as detailed in section F.10, although complete elimination of the curl remains a challenge. In summary, PIWSL minimizes individual energy and force errors, improving the overall accuracy of MLIPs.

To further assess PIWSL's impact, we evaluate the robustness during MD simulations of the MLIP models trained with and without PIWSL. We consider MD simulations of the aspirin molecule, with corresponding results presented in figure 3. We measured stability following the approach proposed in Fu et al. (2023). A detailed experimental setup is provided in section F.4.2. The results demonstrate that PIWSL improves the stability of MD simulations for both the direct and gradient-based force prediction models. The simulation times in figure 3 are shorter than those reported by Fu et al. (2023). This difference arises from our choice to perform MD simulations in the microcanonical (NVE) statistical ensemble instead of the canonical (NVT) statistical ensemble in Fu et al. (2023) to assess stability more accurately without the influence of a thermostat. Results for MD simulations conducted in the canonical statistical ensemble are provided in section F.4.2.

## 5.4 TRAINING MLIPs WITHOUT REFERENCE FORCES

In the following, we explore scenarios where only potential energy labels are available. This situation commonly arises when calculating energy labels with chemically accurate approaches, such as CCSD(T)/CBS (Hobza & Šponer, 2002; Feller et al., 2006), for which force calculation is infeasible. To consider practical applications, we examine two cases: (1) predicting force by a force branch (FB) and (2) predicting force as a gradient of the potential energy (GF). The former enables fast force prediction and is popular in the machine learning community, while the latter requires additional gradient calculation but yields curl-free force predictions. It is popular in computational chemistry as it ensures the conservation of the total energy during MD simulations. The results are provided in Table 3; training without reference forces is achieved by setting the relative force contribution to zero in Eq. (1). The PIWSL method consistently performs better than the baseline for the FB and GF cases. However, a more significant improvement in the force prediction performance is observed in the GF case. We attribute this phenomenon to the inherent nature of PIWSL, which requires consistency between the potential energy and atomic forces, as discussed in section 5.3. This result aligns with our expectations, confirming the capability of our PIWSL method to enable ML models to reduce

the error in the predicted forces. Overall, PIWSL opens a new possibility for training MLIP models using highly accurate reference methods, such as CCSD(T)/CBS.

Table 3: **Results for models trained on the ANI-1x data set without reference forces.** All models are trained using 1000 training samples. FB refers to the setting where the force branch estimates the force, and GF denotes the setting where the force is estimated by the gradient of the potential energy with respect to the atomic coordinates.

| Model | Case | | Baseline | PIWSL |
|---|---|---|---|---|
| PaiNN | FB | E | $42.36 \pm 0.30$ | $\mathbf{25.42 \pm 0.72}$ |
| | | F | $24.25 \pm 0.00$ | $\mathbf{20.54 \pm 0.08}$ |
| | GF | E | $41.83 \pm 1.81$ | $\mathbf{29.71 \pm 0.55}$ |
| | | F | $83.36 \pm 2.85$ | $\mathbf{24.02 \pm 0.95}$ |
| Equiformer | FB | E | $43.14 \pm 0.86$ | $\mathbf{29.48 \pm 0.51}$ |
| | | F | $24.25 \pm 0.00$ | $\mathbf{21.99 \pm 0.49}$ |
| | GF | E | $42.55 \pm 0.99$ | $\mathbf{32.66 \pm 1.11}$ |
| | | F | $35.70 \pm 0.78$ | $\mathbf{21.83 \pm 0.27}$ |

Table 4: **Results for models trained on the ANI-1x data set with ablated weakly supervised losses.** All models are trained using 1000 training samples. All results are obtained by averaging over three independent runs. Energy RMSE is given in kcal/mol, while force RMSE is given in kcal/mol/Å.

| Model | PITC | PISC | E | F |
|---|---|---|---|---|
| PaiNN | ✗ | ✗ | $56.62 \pm 2.80$ | $12.96 \pm 0.06$ |
| | ✓ | ✗ | $\mathbf{24.60 \pm 0.18}$ | $\mathbf{11.51 \pm 0.03}$ |
| | ✗ | ✓ | $58.30 \pm 2.10$ | $13.18 \pm 0.29$ |
| | ✓ | ✓ | $\mathbf{24.53 \pm 0.48}$ | $\mathbf{11.43 \pm 0.05}$ |
| Equiformer | ✗ | ✗ | $54.52 \pm 4.52$ | $10.10 \pm 0.00$ |
| | ✓ | ✗ | $32.64 \pm 26.48$ | $\mathbf{9.64 \pm 0.03}$ |
| | ✗ | ✓ | $48.96 \pm 4.96$ | $10.30 \pm 0.06$ |
| | ✓ | ✓ | $\mathbf{20.89 \pm 0.50}$ | $\mathbf{9.68 \pm 0.03}$ |

## 5.5 FURTHER ANALYSES OF PIWSL

The following provides further analyses of our approach. We provide the results for Equiformer v2 and PaiNN since these models employ equivariant features and demonstrate a high accuracy on the ANI-1x data set when trained using 1000 configurations.

**Comparing PITC with the Taylor-Expansion-Based Weak Label Loss.** We compare the PIWSL method with the Taylor-expansion-based weak label (WL) approach (Cooper et al., 2020), whose loss function is presented in Eq. (A3). For simplicity, we only consider the PITC loss in Eq. (3). For a fair comparison, we consider the following two cases. First, we train with reference forces and energies (w. RF). Second, we train the methods without reference forces and use only the reference energies. For the training with reference forces, we set the numeric coefficient of the PITC loss to $1.0$; for the training without reference forces, the coefficient is set to $0.1$. Note that the WL loss without the reference force is calculated using the predicted force labels. The results are provided in Table 5.

Our PITC loss demonstrates the best accuracy in all cases, with and without the reference forces. Interestingly, PaiNN failed to learn the potential energy with the WL loss and reference forces. We hypothesize this to be due to the imbalance of the training between the energies and forces. Specifically, the WL loss trains only the potential energy, resulting in an inconsistency between the energy and force branches, which share the same readout layer that experiences more frequent updates using the potential energy. This hypothesis is supported by the results for the training without reference forces, where the error in energy is reduced compared to the baseline. A further validation in a similar experiment in the case of GF is provided in section F.9. However, the proposed PITC loss still performs better here. In summary, the PITC loss enables MLIPs to learn energies and forces consistent with each other and does it better than the previously proposed WL method.

**Ablating the Impact of PITC and PISC Losses.** We conduct an ablation experiment to analyze the impact of PITC and PISC losses. Results in Table 4 indicate that the PITC loss predominantly improves the accuracy of resulting models, especially for PaiNN. Using just the PISC loss does not consistently improve accuracy but stabilizes training when combined with PITC. This combined approach notably benefits Equiformer v2. For Equiformer v2, we repeated the experiment five times to reduce the effect from an outlier on the PITC loss.

**Adversarial Directions for Perturbing Atomic Positions.** The following discusses the dependence of the PIWSL's performance on selecting the vector $\delta \mathbf{r}$ in Eq. (5) employed to perturb atomic positions. The detailed implementation and setups are provided in section D.1. Table 6 compares the results obtained for a randomly-sampled vector $\delta \mathbf{r}$ and for the one determined adversarially. The results demonstrate that both approaches improve the performance compared to the baseline without weak supervision, though the results might depend on the employed model.

Table 5: **Comparison of PITC and the Taylor-expansion-based weak label loss.** WL (+FP) denotes the Taylor-expansion-based method using reference energies and either reference (w. RF) or predicted (w/o. RF) forces; see Eq. (A3). The listed values are the RMSE values for energies in kcal/mol and atomic forces in kcal/mol/Å. All models are trained on the ANI-1x data set using 1000 configurations, with (w.) and without (w/o.) reference atomic forces (RF).

| Model | Case | | Baseline | PITC | WL (+FP) |
|---|---|---|---|---|---|
| PaiNN | (w. RF) | E | $56.62 \pm 2.80$ | $\mathbf{30.94 \pm 0.56}$ | $81.86 \pm 9.39$ |
| | | F | $12.96 \pm 0.06$ | $\mathbf{12.04 \pm 0.04}$ | $14.54 \pm 0.12$ |
| | (w/o. RF) | E | $42.36 \pm 0.30$ | $\mathbf{25.42 \pm 0.72}$ | $41.77 \pm 4.82$ |
| | | F | $24.25 \pm 0.00$ | $\mathbf{20.54 \pm 0.08}$ | $24.68 \pm 0.54$ |
| Equiformer | (w. RF) | E | $54.52 \pm 4.52$ | $\mathbf{23.16 \pm 0.19}$ | $31.02 \pm 3.99$ |
| | | F | $10.10 \pm 0.00$ | $\mathbf{10.03 \pm 0.05}$ | $13.43 \pm 0.92$ |
| | (w/o. RF) | E | $43.14 \pm 0.86$ | $\mathbf{29.48 \pm 0.51}$ | $88.59 \pm 11.36$ |
| | | F | $24.25 \pm 0.00$ | $\mathbf{21.99 \pm 0.49}$ | $293.41 \pm 26.96$ |

Table 6: **PIWSL's performance dependence on the atomic position perturbation vector.** The numerical values are RMSEs for the energy in kcal/mol and force in kcal/mol/Å. All results are provided for the ANI-1x data set and models trained using 1000 configurations.

| | | Baseline | Random (Eq. (6)) | Adversarial (Eq. (7)) |
|---|---|---|---|---|
| PaiNN | E | $56.62 \pm 2.80$ | $\mathbf{24.53 \pm 0.48}$ | $33.67 \pm 1.12$ |
| | F | $12.96 \pm 0.18$ | $\mathbf{11.43 \pm 0.05}$ | $12.74 \pm 0.14$ |
| Equiformer | E | $54.52 \pm 4.52$ | $23.16 \pm 0.50$ | $\mathbf{20.54 \pm 0.21}$ |
| | F | $10.10 \pm 0.00$ | $10.03 \pm 0.03$ | $\mathbf{9.93 \pm 0.04}$ |

# 6 DISCUSSION AND LIMITATIONS

This work introduces the PIWSL method, encompassing two distinct physics-informed weakly supervised loss functions, for learning MLIPs. These losses provide the physics-informed weak labels based on the Taylor expansion (PITC loss) and the spatial consistency (PISC loss) of the potential energy. These physics-informed weak labels enable any MLIP to improve its accuracy and robustness, particularly in scenarios characterized by sparse training data, which are common when investigating a new molecular or material system. The improved accuracy and robustness of MLIPs can allow running sufficiently long MD simulations, resulting in a more effective use of active learning approaches. Our extensive experiments demonstrate notable efficacy and efficiency of our method from various aspects: (i) dependence on the training data set size, (ii) the potential energy prediction variance and robustness in terms of a perturbation on a C–H bond length as well as robustness during MD simulations, and (iii) selection of the perturbation vector. In particular, it is shown that our PIWSL method enables ML models to improve the force prediction even without force labels, thereby opening a new possibility for training MLIPs using highly accurate reference methods, such as CCSD(T)/CBS.

**Limitations.** The proposed PIWSL method is tailored to ML models that predict atomic forces and total energies of atomic systems. It cannot be applied to other ML problems unrelated to computational chemistry or materials science applications. Although this work uses the first-order Taylor expansion to obtain weak labels in Eq. (2), employing a more sophisticated higher-order ordinary differential equation solver is a viable alternative.

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

## A  APPENDIX

## B  POTENTIAL BROADER IMPACT AND ETHICAL ASPECTS

This paper presents work whose goal is to advance the field of machine learning, in particular, machine learning for science. Due to the generic nature of pure science, there are many potential societal consequences of our work in the far future, none of which we feel must be specifically highlighted here.

## C  RELATED WORK

**Addressing Data Sparsity in MLIPs.** Generating training data sets suitable for learning reliable MLIPs is challenging, especially when considering unexplored molecular and material systems. Numerous computationally expensive calculations with either ab initio or first-principles approaches are required for the latter. To mitigate this challenge, active learning (AL) methods, which utilize prediction uncertainty, can be applied (Li et al., 2015; Podryabinkin & Shapeev, 2017; Vandermause et al., 2020; Shuaibi et al., 2021b; Briganti & Lunghi, 2023; Zaverkin et al., 2024b). Furthermore, equivariant MLIPs often reduce required training data set sizes through improved data efficiency (Batzner et al., 2022; Batatia et al., 2022).

## D  EXPERIMENTAL SETUP, BASELINES, AND DATA SETS

### D.1  EXPERIMENTAL SETUP

**Code for Experiments.** The code used to run our experiments builds upon the recent work (Fu et al., 2023) and extends it to integrate the latest Open Catalyst Project code (Chanussot et al., 2021). We adopt hyper-parameters from the Open Catalyst (OC) project, tuned to the corresponding OC data set. Note that we do not use this data set in the presented work, whose main focus is training general-purpose MLIPs that can be used to run molecular dynamics (MD) simulations and geometry optimization. However, the OC data set has been designed to investigate the latter, making it less suitable for the current study. Our modifications include adjusting the learning rate scheduler, details of which can be found in our repository. For potential energy and force prediction, we utilize mean-absolute error (MAE) and $L_2$-norm (L2MAE) losses with coefficients of 1 and 100, respectively. More details on the model hyperparameters are provided in our repository. For the PITC and PISC loss functions, we use the mean square error (MSE) loss based on an experiment in section F.5.

**Training Details.** For training MLIPs, we followed the setup in the Open Catalyst Project. We kept the mini-batch size consistent across all models, as shown in Table A1. We have chosen the mini-batch size based on the maximum memory needed by the most demanding models, such as eSCN and Equiformer v2. All experiments are performed on a single NVIDIA A100 GPU with 81.92 GB memory. To avoid overfitting, we stopped training when the validation loss stopped improving—the specific number of training iterations is provided in Table A2.

We used perturbation vectors $\delta\mathbf{r}$ drawn from a uniform random distribution; see also section 4.4. Particularly, we defined $\delta\mathbf{r} \equiv \epsilon\mathbf{g}$, with each component of $\mathbf{g}$ drawn from a uniform random distribution in the interval $(-1, 1)$. The magnitude $\epsilon$ is also drawn from a uniform random distribution and

Table A1: **Employed mini-batch sizes.** We provide mini-batch sized for all data sets and models employed in this work.

|                 | ANI-1x | TiO$_2$$^a$ | rMD17 | LMNTO |
| --------------- | ------ | ----------- | ----- | ----- |
| Mini-batch size | 6      | 4           | 16    | 4     |

$^a$ As explained in Table 2, the mini-batch size of the SchNet model was changed to 32 due to high RMSE values observed with a mini-batch size of four as the training data set size increased. A more detailed discussion of the results for SchNet is provided in section F.1.

Table A2: **Total number of training iterations.** We provide the total number of training iterations for all data sets and training set sizes employed in this work. The number in the parentheses demonstrates the corresponding total number of training epochs.

| $N_{\text{train}}$ | ANI-1x | TiO2 | rMD17 | LMNTO |
|---:|---|---|---|---|
| 50 | 7500 (900) | – | – | – |
| 100 | 10,000 (600) | 10,000 (400) | 7500 (1200) | 10,000 (400) |
| 1000 | 40,000 (240) | 10,000 (100) | 10,000 (160) | 10,000 (100) |
| 10,000 | 100,000 (60) | – | – | – |
| 100,000 | 400,000 (30) | – | – | – |
| 1,000,000 | 400,000 (3) | – | – | – |
| 5,000,000 | 420,000 (1) | – | – | – |

Table A3: **Hyper-parameters for the PIWSL loss.** We selected the following hyper-parameter combinations using Optuna (Akiba et al., 2019): $(C_{\text{PITC}}, C_{\text{PISC}}, \epsilon_{\max})$= Case A: (1.2, 0.8, 0.025), Case B: (1.0, 0, 0.01), Case C: (0.1, 0.01, 0.01), Case D: (1.2, 0.01, 0.025), Case E: (1.2, 0.01, 0.01), Case F: (1.2, 0.01, 0.015), Case G: (0.01, 0.001, 0.025), and Case H: (0.1, 0.01, 0.025).

| Dataset | Size | Equiformer v2 | eSCN | PaiNN | SpinConv | SchNet |
|---|---:|---|---|---|---|---|
| ANI-1x | 50 | A | C | B | A | A |
| | 100 | A | C | A | D | A |
| | 1000 | D | D | D | B | B |
| | 10,000 | G | C | B | C | C |
| | 100,000 | – | – | C | – | – |
| | 1,000,000 | – | – | C | – | – |
| | 5,000,000 | – | – | C (E) | – | – |
| TiO2 | 100 | A | A | A | A | H |
| | 1000 | G | A | C | A | C |
| rMD17 (Aspirin) | 100 | E | B | F | D | A |
| | 1000 | B | B | B | B | B |
| rMD17 (Benzene) | 100 | B | B | B | B | B |
| | 1000 | G | B | B | B | B |
| rMD17 (Naphthalene) | 100 | G | B | B | B | B |
| | 1000 | G | B | B | B | B |
| LMNTO | 100 | B | B | B | A | B |
| | 1000 | B | A | B | B | B |

$\epsilon < \epsilon_{\max}$. This definition of $\delta\mathbf{r}$ differs from the one in Eq. (6), improving the computational efficiency of PIWSL by avoiding the calculation of square root and division.

The remaining hyper-parameters are the coefficients for the PITC and PISC losses $(C_{\text{PITC}}, C_{\text{PISC}})$ and the maximum magnitude $\epsilon_{\max}$ of the perturbation vector $\delta\mathbf{r}$; see Table A3. These hyper-parameters are tuned using Optuna (Akiba et al., 2019) for PaiNN and Equiformer v2. We used 1000 configurations drawn randomly from the original ANI-1x data set for training. Due to multiple local minima, Optuna identified several optimal hyper-parameter sets in each run. We selected the following representative combinations $(C_{\text{PITC}}, C_{\text{PISC}}, \epsilon_{\max})$ = Case A: (1.2, 0.8, 0.025), Case B: (1.0, 0, 0.01), Case C: (0.1, 0.01, 0.01), Case D: (1.2, 0.01, 0.025), Case E: (1.2, 0.01, 0.01), Case F: (1.2, 0.01, 0.015), Case G: (0.01, 0.001, 0.025), and Case H: (0.1, 0.01, 0.025). We selected the hyper-parameters listed in Table A3 based on the validation dataset performance.

**Splitting Data Sets.** We split the original data sets into training, validation, and test sets for our experiments. We shuffled the original data sets using a random seed and selected the training data sets of predefined sizes. For validation, we selected the same number of configurations as in the training data set if it exceeded 100 configurations; otherwise, we used 100 configurations to ensure sufficient validation size. For the rMD17 data set, following (Fu et al., 2023), we used 9000 configurations as

a validation data set and another 10,000 for testing. We used the same test data set across different sizes of the training data sets for a fair performance comparison. We used 10,000 test configurations for ANI-1x and 1000 for $TiO_2$ and LMNTO.

**Training with Adversarial Directions.** In our experiments, which defined perturbation vectors adversarially (see section 5.5), we determined adversarial directions using Eq. (7). More concretely, we only considered the potential energy, i.e., $\mathbf{y}_{\text{pred}}$ and $\mathbf{y}_{\text{label}}$, to avoid Hessian calculations. In addition, we considered the loss function for the potential energy as $L_{\text{dist}}$ in Eq. (7). The expression of $\mathbf{g} = \nabla_{\mathbf{r}} L_{\text{dist}}$ is then

$$\mathbf{g}_{\mathcal{S}} = \nabla_{\mathbf{r}_i} \sqrt{(E(\mathcal{S};\boldsymbol{\theta}) - E_{\mathcal{S}}^{\text{ref}})^2} = -\frac{1}{2L_{\text{dist}}}(\mathbf{F}_i(\mathcal{S};\boldsymbol{\theta}) - \mathbf{F}_{i,\mathcal{S}}^{\text{ref}})(E(\mathcal{S};\boldsymbol{\theta}) - E_{\mathcal{S}}^{\text{ref}}). \tag{A1}$$

Note that we used the relation $\nabla_{\mathbf{r}_i} E_{\mathcal{S}}^{\text{ref}} = -\mathbf{F}_{i,\mathcal{S}}^{\text{ref}}$ to obtain the final expression. Though $E^{\text{ref}}$ can also be interpreted as a constant regarding atom positions. We have chosen this expression to avoid the case where the adversarial direction $\mathbf{g}$ points to $\mathbf{F}_{i,\mathcal{S}}^{\text{ref}}$ other than the very beginning of the training. Our experiments indicate that the employed expression is slightly better than its alternative. In our experiment, we also randomly flip the sign of $\mathbf{g}_{\mathcal{S}}$ to avoid overfitting to adversarial directions.

## D.2 BASELINE METHODS

**Data Augmentation with NoisyNode.** In our experiments, we used the NoisyNode approach (Godwin et al., 2022) as one of the baseline methods. This method aims to improve the performance of ML models by adding a perturbation to node features, i.e., atomic coordinates, and makes ML models recover original labels. This approach enables ML models to be more robust to noise in the data. Although the original method recommends adding a decoder network to learn the denoising process, we do not utilize it following previous work (Liao et al., 2023) and add the perturbation vector to atomic coordinates similar to PIWSL losses, fixing energy and force labels. We implement the NoisyNode approach in our code. Thus, we can expect slightly different behavior compared to the recent work (Godwin et al., 2022; Liao et al., 2023) [2].

**Taylor-Expansion-Based Weak Labels.** Recent work proposed a similar Taylor-expansion-based weak label approach (Cooper et al., 2020). Nonetheless, the loss is different from the one in Eq. (3) as the authors used reference energy and atomic force labels to estimate weak energy labels $E_{\mathcal{S}_{\delta\mathbf{r}}}^{\text{ref}}$ for perturbed atomic configurations $\mathcal{S}_{\delta\mathbf{r}}$

$$E_{\mathcal{S}_{\delta\mathbf{r}}}^{\text{ref}} \approx E_{\mathcal{S}}^{\text{ref}} - \sum_{i=1}^{N_{\text{at}}} \left\langle \delta\mathbf{r}_i, \mathbf{F}_{i,\mathcal{S}}^{\text{ref}} \right\rangle + \mathcal{O}\left(||\delta\mathbf{r}||^2\right). \tag{A2}$$

The trainable parameters of MLIPs are optimized by minimizing the weak label (WL) loss

$$L_{\text{WL}}(\mathcal{S};\boldsymbol{\theta}) = \ell\left(E(\mathcal{S}_{\delta\mathbf{r}};\boldsymbol{\theta}), E_{\mathcal{S}}^{\text{ref}} - \sum_{i=1}^{N_{\text{at}}} \left\langle \delta\mathbf{r}_i, \mathbf{F}_{i,\mathcal{S}}^{\text{ref}} \right\rangle\right). \tag{A3}$$

Figure 1 (a) illustrates the corresponding approach (Cooper et al., 2020), which computes the energy of a perturbed atomic configuration using a Taylor expansion based on reference energy and atomic force labels. This approach was originally applied to train MLIPs without explicit force labels.

## D.3 DESCRIPTION OF THE DATA SETS

**ANI-1x Data Set.** The ANI-1x data set is a heterogeneous molecular data set and includes 63,865 organic molecules (with chemical elements H, C, N, and O) whose size ranges from 4 to 64 atoms (Smith et al., 2020). The ML model requires learning total energies and atomic forces for various molecules and their conformations. Total energies and atomic forces are obtained through DFT calculations.

**$TiO_2$ Data Set.** Titanium dioxide ($TiO_2$) is an industrially relevant and well-studied material. $TiO_2$ dataset includes 7815 bulk structures of several $TiO_2$ phases whose reference energies and forces

---

[2] Note that NoisyNode assumes the unperturbed state to be the equilibrium structure, which may have contributed to the limited performance improvements observed in our experiments. Recently, efforts have been underway to develop an extended version of NoisyNode tailored for non-equilibrium structures, such as (Liao et al., 2024).

are obtained through DFT calculations (Artrith & Urban, 2016). The number of atoms in a single configuration ranges from 6 to 95.

**rMD17 Data Set.** The rMD17 data set includes ten small organic molecules, including 100,000 configurations obtained by running MD simulations for each (Christensen & von Lilienfeld, 2020). The ML model requires learning the total energies and atomic forces for each molecule. In this revised version of the MD17 data set, the molecules are taken from the original MD17 data set (Chmiela et al., 2017; 2018). However, the energies and forces are recalculated at the PBE/def2-SVP level of theory using very tight SCF convergence and a very dense DFT integration grid.

**LMNTO Data Set.** The Li-Mo-Ni-Ti oxide (LMNTO) is of technological significance as a potential high-capacity positive electrode material for lithium-ion batteries. It exhibits substitutional disorder, with Li, Mo, Ni, and Ti all sharing the same sublattice. This data set includes LMNTO with the composition $Li_8Mo_2Ni_7Ti_7O_{32}$ and configurations obtained from an MD simulation, resulting in approximately 2600 structures in total (Cooper et al., 2020).

**MD22 Data Set.** The MD22 data set (Chmiela et al., 2023) includes seven larger organic molecules, such as a small peptide and a double-walled nanotube, whose atom number ranges from 42 to 370. The data set consists of MD trajectories sampled at temperatures between 400 and 500 K. The ML model requires learning the total energies and atomic forces for each molecule. The energies and forces are calculated at the PBE+MBD level of theory.

## E DIFFERENCES IN GRADIENTS FOR PHYSICS-INFORMED LOSSES

The following considers the gradients of the proposed two losses. First, considering squared errors, we obtain the following gradients of the loss in Eq. (A2) with respect to trainable parameters

$$\frac{\mathrm{d}\mathcal{L}_{\mathrm{WL}}}{\mathrm{d}\boldsymbol{\theta}} = 2\left(E\left(\mathcal{S}_{\delta\mathbf{r}};\boldsymbol{\theta}\right) - E_{\mathcal{S}}^{\mathrm{ref}} + \sum_{i=1}^{N_{\mathrm{at}}}\langle\delta\mathbf{r}_i, \mathbf{F}_{i,\mathcal{S}}^{\mathrm{ref}}\rangle\right)\frac{\mathrm{d}E\left(\mathcal{S}_{\delta\mathbf{r}};\boldsymbol{\theta}\right)}{\mathrm{d}\boldsymbol{\theta}}. \tag{A4}$$

In contrast, for the PITC loss in Eq. (3) we obtain

$$\begin{aligned}\frac{\mathrm{d}\mathcal{L}_{\mathrm{PITC}}}{\mathrm{d}\boldsymbol{\theta}} =&2\left(E\left(\mathcal{S}_{\delta\mathbf{r}};\boldsymbol{\theta}\right) - E\left(\mathcal{S};\boldsymbol{\theta}\right) + \sum_{i=1}^{N_{\mathrm{at}}}\langle\delta\mathbf{r}_i, \mathbf{F}_i\left(\mathcal{S};\boldsymbol{\theta}\right)\rangle\right)\times\\&\left(\frac{\mathrm{d}E\left(\mathcal{S}_{\delta\mathbf{r}};\boldsymbol{\theta}\right)}{\mathrm{d}\boldsymbol{\theta}} - \frac{\mathrm{d}E\left(\mathcal{S};\boldsymbol{\theta}\right)}{\mathrm{d}\boldsymbol{\theta}} + \sum_{i=1}^{N_{\mathrm{at}}}\frac{\mathrm{d}\langle\delta\mathbf{r}_i, \mathbf{F}_i\left(\mathcal{S};\boldsymbol{\theta}\right)\rangle}{\mathrm{d}\boldsymbol{\theta}}\right).\end{aligned} \tag{A5}$$

The above equations indicate that the direction of the derivative of the PITC loss in Eq. (A5) is different from that of the weak label loss because of the incorporation of the predicted potential energy at the original and the force at the reference point. The gradient of PISC loss in Eq. (4) reads

$$\begin{aligned}\frac{\mathrm{d}\mathcal{L}_{\mathrm{PISC}}}{\mathrm{d}\boldsymbol{\theta}} = &2\left(E\left(\mathcal{S};\boldsymbol{\theta}\right) - \sum_{i=1}^{N_{\mathrm{at}}}\langle\delta\mathbf{r}_i, \mathbf{F}_i\left(\mathcal{S};\boldsymbol{\theta}\right)\rangle - E\left(\mathcal{S}_{\delta\mathbf{r}'};\boldsymbol{\theta}\right) + \sum_{i=1}^{N_{\mathrm{at}}}\langle\delta\mathbf{r}_i'', \mathbf{F}_i\left(\mathcal{S}_{\delta\mathbf{r}'};\boldsymbol{\theta}\right)\rangle\right)\\&\times\left(\frac{\mathrm{d}E\left(\mathcal{S};\boldsymbol{\theta}\right)}{\mathrm{d}\boldsymbol{\theta}} - \sum_{i=1}^{N_{\mathrm{at}}}\frac{\mathrm{d}\langle\delta\mathbf{r}_i, \mathbf{F}_i\left(\mathcal{S};\boldsymbol{\theta}\right)\rangle}{\mathrm{d}\boldsymbol{\theta}} - \frac{\mathrm{d}E\left(\mathcal{S}_{\delta\mathbf{r}'};\boldsymbol{\theta}\right)}{\mathrm{d}\boldsymbol{\theta}} + \sum_{i=1}^{N_{\mathrm{at}}}\frac{\mathrm{d}\langle\delta\mathbf{r}_i'', \mathbf{F}_i\left(\mathcal{S}_{\delta\mathbf{r}'};\boldsymbol{\theta}\right)\rangle}{\mathrm{d}\boldsymbol{\theta}}\right).\end{aligned} \tag{A6}$$

## F EXPERIMENTS

### F.1 BENCHMARK RESULTS

The following section provides additional results, complementing those provided in the main text.

**Additional Results for ANI-1x.** Table A4 provides results for ANI-1x data set and a training set sizes of 50 or 10,000; see also figure 2. The table demonstrates a considerable reduction of energy and force RMSEs for models trained using small training data set sizes of 50 configurations. Furthermore, we find around 5 to 25 % error reduction for a larger training set size of 10,000, indicating the

Table A4: **Energy (E) and force (F) root-mean-square errors (RMSEs) for the ANI-1x data set.** The results are obtained by averaging over three independent runs. Energy RMSE is given in kcal/mol, while force RMSE is in kcal/mol/Å.

| | | $N_{\text{train}} = 50$ | | | $N_{\text{train}} = 10,000$ | | |
| | | Baseline | Noisy Nodes | PIWSL | Baseline | Noisy Nodes | PIWSL |
| --- | --- | --- | --- | --- | --- | --- | --- |
| Schnet | E | $90.08 \pm 1.24$ | $\mathbf{76.83 \pm 0.75}$ | $83.90 \pm 2.82$ | $24.88 \pm 0.01$ | $\mathbf{24.86 \pm 0.00}$ | $24.88 \pm 0.00$ |
| | F | $35.49 \pm 0.36$ | $\mathbf{31.13 \pm 0.13}$ | $35.30 \pm 0.87$ | $13.36 \pm 0.01$ | $13.36 \pm 0.00$ | $13.36 \pm 0.00$ |
| PaiNN | E | $212.64 \pm 1.14$ | $440.11 \pm 11.68$ | $\mathbf{121.36 \pm 4.13}$ | $19.14 \pm 0.38$ | $165.25 \pm 4.87$ | $\mathbf{14.10 \pm 0.14}$ |
| | F | $22.61 \pm 0.04$ | $22.50 \pm 0.22$ | $\mathbf{20.83 \pm 0.28}$ | $8.24 \pm 0.10$ | $9.22 \pm 0.09$ | $\mathbf{7.89 \pm 0.02}$ |
| SpinConv | E | $222.75 \pm 7.12$ | $219.85 \pm 6.99$ | $\mathbf{175.38 \pm 9.77}$ | $19.42 \pm 0.67$ | $46.31 \pm 10.31$ | $\mathbf{18.81 \pm 0.60}$ |
| | F | $\mathbf{24.88 \pm 0.88}$ | $24.61 \pm 0.35$ | $25.12 \pm 0.58$ | $10.31 \pm 0.33$ | $10.78 \pm 0.66$ | $\mathbf{9.94 \pm 0.12}$ |
| eSCN | E | $517.17 \pm 31.98$ | $583.90 \pm 33.04$ | $\mathbf{454.40 \pm 11.10}$ | $12.65 \pm 0.63$ | $165.30 \pm 33.11$ | $\mathbf{10.66 \pm 0.31}$ |
| | F | $22.51 \pm 0.09$ | $24.04 \pm 0.15$ | $\mathbf{22.28 \pm 0.08}$ | $5.11 \pm 0.30$ | $11.51 \pm 0.23$ | $\mathbf{4.35 \pm 0.15}$ |
| Equiformer | E | $498.58 \pm 17.44$ | $630.32 \pm 0.32$ | $\mathbf{433.88 \pm 79.63}$ | $8.03 \pm 0.21$ | $970.95 \pm 236.90$ | $\mathbf{7.77 \pm 0.14}$ |
| | F | $22.86 \pm 0.04$ | $22.92 \pm 0.00$ | $\mathbf{22.72 \pm 0.04}$ | $\mathbf{2.97 \pm 0.00}$ | $29.28 \pm 5.63$ | $2.98 \pm 0.00$ |

Table A5: **Energy and force erorrs of PaiNN model trained on the ANI-1x data set with 100,000, 1,000,000, and 5,000,000 samples.** The results are obtained by averaging over three independent runs. Energy errors are given in kcal/mol, while force errors are in kcal/mol/Å.

| | $N_{\text{train}}$ | Force MAE | Force RMSE | Energy MAE | Energy RMSE |
| --- | --- | --- | --- | --- | --- |
| Baseline | 100,000 | $\mathbf{0.92 \pm 0.00}$ | $\mathbf{3.70 \pm 0.01}$ | $4.28 \pm 0.15$ | $6.14 \pm 0.21$ |
| PIWSL | | $0.91 \pm 0.01$ | $3.72 \pm 0.04$ | $\mathbf{4.07 \pm 0.17}$ | $\mathbf{5.83 \pm 0.20}$ |
| Baseline | 1,000,000 | $\mathbf{0.67 \pm 0.00}$ | $\mathbf{2.74 \pm 0.01}$ | $4.90 \pm 0.23$ | $6.56 \pm 0.26$ |
| PIWSL | | $0.68 \pm 0.00$ | $2.77 \pm 0.04$ | $\mathbf{4.48 \pm 0.05}$ | $\mathbf{6.06 \pm 0.01}$ |
| Baseline | $5,000,000^{a}$ | 0.53 | 2.18 | 3.94 | 5.24 |
| PIWSL (C)$^{b}$ | | **0.52** | **2.16** | 3.85 | 5.18 |
| PIWSL (E)$^{b}$ | | 0.55 | 2.24 | **3.45** | **4.92** |

[a] Because of the computational cost, we performed only one training in the case of 5,000,000 training samples.
[b] This symbol denote the hyper-parameter set defined in Table A3

effectiveness of the PIWSL method for relatively large training set sizes. Finally, we provide a result of PaiNN model trained on ANI-1x data set with 100,000, 1,000,000 and 5,000,000 samples in Table A5, which demonstrates that PIWSL still improves the performance around 5% to 10% of the energy RMSE.

**Additional Results for SchNet Applied to $TiO_2$.** In Table 2, we set the mini-batch size to 32 for training the SchNet model. This adjustment was made because training SchNet with a small mini-batch size of four increases RMSE values with a growing training data set size. Table A6 demonstrates the performance of the SchNet model for a mini-batch size of four. Table A7 provides the results obtained for the SchNet model with a mini-batch size of four for the following training set sizes: 100, 200, 500, and 1000. This figure demonstrates that SchNet, with a mini-batch size of four, reaches its best performed with $N_{\text{train}} = 200$. These results indicate the difficulty of learning training data statistics from small mini-batches, probably due to the limited expressive power of SchNet.

**Results for LMNTO.** Table A8 presents RMSE errors for LMNTO (Cooper et al., 2020). PIWSL shows the error reduction for most cases for this benchmark data set, especially for small training set sizes (i.e., a training set of 100 configurations).

**Molecular Dynamics Trajectories (rMD17).** Table A9 analyzes the effect of the PIWSL on data sets containing conformations of a single small molecule, different from the heterogeneous ANI-1x data set. For this purpose, we have chosen the benzene ($n_{\text{atom}} = 12$), naphthalene ($n_{\text{atom}} = 18$), and aspirin ($n_{\text{atom}} = 21$) molecules because these represent molecules of different sizes. The results in Table A9 indicate that our approach is still effective in this scenario. However, the PIWSL

Table A6: **Energy (F) and force (F) root-mean-square errors (RMSEs) for the TiO2 data set obtained for the SchNet model with a mini-batch size of four.** The results are obtained by averaging over three independent runs. Energy RMSE is given in kcal/mol, while force RMSE is in kcal/mol/Å.

| | | $N_{\text{train}} = 100$ | | | $N_{\text{train}} = 1000$ | | |
| | | Baseline | Noisy Nodes | PIWSL | Baseline | Noisy Nodes | PIWSL |
|---|---|---|---|---|---|---|---|
| SchNet | E | $18.85 \pm 0.00$ | $\mathbf{17.48 \pm 0.00}$ | $17.58 \pm 0.00$ | $35.58 \pm 0.00$ | $58.08 \pm 18.44$ | $\mathbf{15.28 \pm 0.12}$ |
| | F | $2.74 \pm 0.00$ | $\mathbf{2.51 \pm 0.00}$ | $2.74 \pm 0.00$ | $6.54 \pm 0.00$ | $18.40 \pm 0.00$ | $\mathbf{3.61 \pm 0.27}$ |

Table A7: **Training data set size dependence of SchNet with a mini-batch size of four.** The results are presented for the TiO$_2$ data set and are obtained by averaging over three independent runs. Energy RMSE is given in kcal/mol, while force RMSE is in kcal/mol/Å.

| | | $N_{\text{train}} = 100$ | $N_{\text{train}} = 200$ | $N_{\text{train}} = 500$ | $N_{\text{train}} = 1000$ |
|---|---|---|---|---|---|
| SchNet | E | $18.85 \pm 0.00$ | $\mathbf{16.28 \pm 0.00}$ | $24.42 \pm 0.00$ | $35.58 \pm 0.00$ |
| | F | $2.74 \pm 0.00$ | $\mathbf{2.56 \pm 0.00}$ | $4.43 \pm 0.00$ | $6.54 \pm 0.00$ |

performance in the case of benzene has nearly no gain. This observation can be attributed to only a small variation of atomic coordinates in the benzene data set, simplifying the learning task. We also note that the obtained results for SchNet and PaiNN are somewhat worse than originally reported (Schütt et al., 2017; Schütt et al., 2021), attributed to the modified implementation in the Open Catalyst project code and the use of the force-branch instead of the gradient-based forces. The effect of the gradient-based forces is investigated in section F.9.

**Molecular Dynamics Trajectories for Large Molecules (MD22).** Table A10 evaluates the impact of PIWSL on data sets containing conformations of a single large molecule. For this purpose, we selected the buckyball catcher molecule ($n_{\text{atom}} = 148$) because of its large size and non-trivial structure. To demonstrate the applicability of PIWSL, we trained a MACE model (Batatia et al., 2022), to prove that PIWSL enhances even the performance of a recent state-of-the-art model. The model structure and training configuration were based on those provided in the official repository[3] with slight modifications: "max_$L$" was set to one and mini-batch size was adjusted to 4. Following the original training setup (Chmiela et al., 2023), we randomly sampled 600 configurations for the

[3]https://mace-docs.readthedocs.io/en/latest/examples/training_examples.html

Table A8: **Energy (E) and force (F) root-mean-square errors (RMSEs) for the LMNTO data set.** The results are obtained by averaging over three independent runs. Energy RMSE is given in kcal/mol, while force RMSE is in kcal/mol/Å.

| Model | | $N_{\text{train}} = 100$ | | | $N_{\text{train}} = 1000$ | | |
| | | Baseline | NoisyNode | PIWSL | Baseline | NoisyNode | PIWSL |
|---|---|---|---|---|---|---|---|
| SchNet | E | $4.46 \pm 0.00$ | $6.10 \pm 0.00$ | $\mathbf{4.45 \pm 0.00}$ | $\mathbf{3.09 \pm 0.00}$ | $3.25 \pm 0.00$ | $\mathbf{3.09 \pm 0.00}$ |
| | F | $9.24 \pm 0.00$ | $\mathbf{8.31 \pm 0.00}$ | $9.24 \pm 0.00$ | $\mathbf{5.09 \pm 0.00}$ | $5.21 \pm 0.00$ | $\mathbf{5.09 \pm 0.00}$ |
| PaiNN | E | $6.91 \pm 0.02$ | $7.09 \pm 0.04$ | $\mathbf{5.99 \pm 0.02}$ | $3.26 \pm 0.01$ | $4.61 \pm 0.03$ | $\mathbf{2.98 \pm 0.01}$ |
| | F | $\mathbf{4.75 \pm 0.00}$ | $7.20 \pm 0.01$ | $\mathbf{4.75 \pm 0.00}$ | $\mathbf{2.03 \pm 0.00}$ | $2.55 \pm 0.00$ | $\mathbf{2.03 \pm 0.00}$ |
| SpinConv | E | $7.90 \pm 0.00$ | $7.83 \pm 0.04$ | $\mathbf{7.83 \pm 0.01}$ | $4.90 \pm 0.33$ | $7.20 \pm 0.06$ | $\mathbf{3.95 \pm 0.02}$ |
| | F | $\mathbf{4.63 \pm 0.01}$ | $5.14 \pm 0.04$ | $4.71 \pm 0.02$ | $1.81 \pm 0.01$ | $2.33 \pm 0.00$ | $\mathbf{1.74 \pm 0.00}$ |
| eSCN | E | $7.92 \pm 0.00$ | $7.92 \pm 0.00$ | $7.92 \pm 0.00$ | $7.93 \pm 0.00$ | $7.93 \pm 0.00$ | $\mathbf{6.40 \pm 0.14}$ |
| | F | $4.67 \pm 0.01$ | $7.59 \pm 0.02$ | $\mathbf{4.64 \pm 0.01}$ | $1.54 \pm 0.00$ | $1.98 \pm 0.06$ | $\mathbf{1.53 \pm 0.00}$ |
| Equiformer v2 | E | $7.40 \pm 0.03$ | $7.92 \pm 0.00$ | $\mathbf{7.32 \pm 0.08}$ | $3.57 \pm 0.05$ | $7.04 \pm 0.03$ | $3.60 \pm 0.02$ |
| | F | $4.26 \pm 0.00$ | $7.60 \pm 0.02$ | $\mathbf{4.24 \pm 0.02}$ | $\mathbf{1.34 \pm 0.00}$ | $1.99 \pm 0.00$ | $\mathbf{1.34 \pm 0.00}$ |

Table A9: **Energy (E) and force (F) root-mean-square errors (RMSEs) for the rMD17 data set.** We have chosen benzene, naphthalene, and aspirin for our experiments. The results are obtained by averaging over three independent runs. Energy RMSE is given in kcal/mol, while force RMSE is in kcal/mol/Å.

| Dataset | Model | | $N_{\text{train}} = 100$ Baseline | NoisyNode | PIWSL | $N_{\text{train}} = 1000$ Baseline | NoisyNode | PIWSL |
|---|---|---|---|---|---|---|---|---|
| Benzene ($n_{\text{atom}} = 12$) | Schnet | E | **0.23 ± 0.00** | 0.58 ±0.00 | **0.23 ±0.00** | **0.17 ± 0.00** | 0.32 ± 0.00 | **0.17 ± 0.00** |
| | | F | **2.32 ± 0.00** | 3.61 ± 0.00 | **2.32 ± 0.00** | **1.27 ± 0.00** | 2.51 ± 0.00 | **1.27 ± 0.00** |
| | PaiNN | E | **0.90 ± 0.02** | 2.29 ± 0.50 | 0.89 ±0.03 | **0.47 ± 0.03** | 0.75 ± 0.02 | 0.49 ± 0.03 |
| | | F | **0.57 ± 0.00** | 5.33 ± 0.16 | **0.57 ± 0.00** | **0.23 ± 0.00** | 2.50 ± 0.00 | 0.30 ± 0.00 |
| | SpinConv | E | 2.27 ± 0.09 | 2.32 ± 0.00 | **1.61 ±0.28** | **0.90 ± 0.12** | 2.35 ± 0.01 | 1.07 ± 0.00 |
| | | F | **0.61 ± 0.01** | 3.56 ± 0.00 | 0.65 ± 0.01 | **0.39 ± 0.00** | 2.33 ± 0.00 | 0.43 ± 0.00 |
| | eSCN | E | **0.59 ± 0.01** | 3.47 ± 0.04 | **0.58 ± 0.03** | 0.20 ± 0.00 | 1.01 ± 0.01 | **0.19 ± 0.00** |
| | | F | **0.74 ± 0.01** | 8.43 ± 0.18 | **0.75 ± 0.02** | **0.14 ± 0.00** | 2.99 ± 0.01 | **0.14 ± 0.00** |
| | Equiformer | E | 1.55 ± 0.01 | 2.08 ± 0.01 | **1.52 ± 0.01** | 0.281 ± 0.01 | 1.68 ± 0.02 | **0.276 ± 0.01** |
| | | F | **0.72 ± 0.00** | 10.32 ± 0.04 | **0.72 ± 0.01** | 0.15 ± 0.00 | 2.89 ± 0.00 | **0.13 ± 0.00** |
| Naphthalene ($n_{\text{atom}} = 18$) | Schnet | E | **1.41 ± 0.00** | 1.92 ± 0.00 | **1.41 ± 0.00** | **1.05 ± 0.00** | 1.49 ± 0.00 | **1.05 ± 0.00** |
| | | F | **5.76 ± 0.00** | 5.96 ± 0.00 | **5.76 ± 0.00** | **3.80 ± 0.00** | 4.08 ± 0.00 | **3.80 ± 0.00** |
| | PaiNN | E | 3.63 ± 0.01 | 5.13 ± 0.06 | **3.54 ± 0.02** | 1.37 ± 0.02 | 2.22 ± 0.04 | **1.33 ± 0.01** |
| | | F | **1.98 ± 0.01** | 10.99 ± 0.05 | 1.99 ± 0.00 | **0.72 ± 0.00** | 2.56 ± 0.01 | **0.72 ± 0.00** |
| | SpinConv | E | 2.96 ± 0.22 | 5.73 ± 0.00 | **2.88 ± 0.02** | **1.80 ± 0.02** | 3.39 ± 0.00 | 2.40 ± 0.18 |
| | | F | 2.04 ± 0.01 | 3.91 ± 0.00 | **1.99 ± 0.01** | 0.97 ± 0.00 | 2.46 ± 0.00 | **0.96 ± 0.00** |
| | eSCN | E | **2.07 ± 0.03** | 7.63 ± 0.05 | 2.12 ± 0.01 | **0.56 ± 0.01** | 2.15 ± 0.29 | 0.58 ± 0.01 |
| | | F | **2.28 ± 0.01** | 9.68 ± 0.23 | 2.32 ± 0.23 | **0.42 ± 0.01** | 2.86 ± 0.03 | **0.42 ± 0.01** |
| | Equiformer | E | 4.37 ± 0.03 | 5.70 ± 0.05 | **4.27 ± 0.01** | **0.71 ± 0.02** | 3.70 ± 0.10 | 0.72 ± 0.02 |
| | | F | 1.93 ± 0.03 | 12.73 ± 0.06 | **1.89 ± 0.00** | 0.43 ± 0.02 | 3.20 ± 0.02 | **0.38 ± 0.06** |
| Aspirin ($n_{\text{atom}} = 21$) | Schnet | E | 3.76 ± 0.00 | **3.56 ± 0.00** | 3.74 ± 0.00 | **2.77 ± 0.00** | 3.08 ± 0.00 | **2.77± 0.00** |
| | | F | 12.32 ± 0.00 | **11.59 ± 0.00** | 12.20 ± 0.00 | **6.63 ± 0.00** | 7.03 ± 0.00 | **6.63 ± 0.00** |
| | PaiNN | E | 6.55 ± 0.03 | 9.36 ± 0.08 | **5.64 ± 0.02** | 4.07 ± 0.01 | 4.10 ± 0.01 | **3.99± 0.01** |
| | | F | **7.38 ± 0.02** | 20.37 ± 0.04 | 7.36 ± 0.03 | 2.17 ± 0.00 | 2.17 ± 0.00 | **2.16 ± 0.01** |
| | SpinConv | E | 5.71 ± 0.04 | 6.11 ± 0.00 | **5.03 ± 0.01** | 4.04 ± 0.09 | 4.12 ± 0.06 | **3.42 ± 0.20** |
| | | F | **8.68 ± 0.03** | 10.17 ± 0.00 | 8.94 ± 0.02 | 1.88 ± 0.00 | 1.89 ± 0.00 | **1.83± 0.01** |
| | eSCN | E | 5.14 ± 0.02 | 6.44 ± 0.04 | **4.82± 0.13** | 1.28 ± 0.03 | 1.28 ± 0.02 | 1.29 ± 0.01 |
| | | F | **6.14 ± 0.03** | 13.88± 0.16 | 6.10 ± 0.03 | 1.30 ± 0.01 | 1.29± 0.02 | 1.30 ± 0.01 |
| | Equiformer | E | 4.79 ± 0.02 | 5.75 ± 0.08 | **4.66 ± 0.04** | 1.83 ± 0.04 | 1.83 ± 0.06 | **1.75 ± 0.01** |
| | | F | **4.86 ± 0.03** | 16.80 ± 0.05 | **4.86 ± 0.03** | 1.00 ± 0.03 | 1.00 ± 0.00 | **0.94 ± 0.08** |

Table A10: **Energy (E) and force (F) mean-absolute errors (MAEs) for the MD22 data set.** We have chosen buckyball-catcher for our experiments. The results are obtained by averaging over three independent runs. Energy MAE is given in kcal/mol/atom, while force MAE is in kcal/mol/Å. Note that "SWA" denotes stochastic weight averaging and potential energy is measured per atom number.

| Dataset | Model | | $N_{\text{train}} = 50$ Baseline | PIWSL | $N_{\text{train}} = 600$ Baseline | PIWSL |
|---|---|---|---|---|---|---|
| Buckyball-Catcher | MACE | E | **0.163 ± 0.015** | **0.155 ± 0.036** | **0.091 ± 0.017** | **0.085 ± 0.005** |
| | | F | **6.780 ± 0.038** | **6.758 ± 0.049** | **1.886 ± 0.011** | 1.899 ± 0.009 |
| | MACE w.t. SWA | E | 0.150 ± 0.014 | **0.106 ± 0.005** | 0.080 ± 0.005 | **0.069 ± 0.002** |
| | | F | **6.844 ± 0.064** | **6.807 ± 0.034** | **2.056 ± 0.079** | 2.052 ± 0.106 |

training dataset, 400 for the validation dataset, and retained the remaining 5102 configurations for testing. To further validate PIWSL's effectiveness in sparse data scenarios, we prepared an additional smaller training dataset comprising only 50 configurations, while keeping the validation dataset size unchanged. The model was trained for 450 and 800 epochs for the 600-sample and 50-sample training datasets, respectively, with stochastic weight averaging (Izmailov et al., 2018) applied over the final 200 epochs. The results presented in Table A10 demonstrate that our approach remains effective on average, particularly in the sparse data regime. In this study, the coefficient of the PITC and PISC losses are set as 0.01 and 0.001 with $\epsilon_{\max} = 0.01$.

## F.2 Training Time Analysis

Table A11: **Training time comparison for experiments with and without PIWSL.** We measure the time required for a single training epoch and provide the results obtained as an average over five epochs. We use 1000 configurations from the ANI-1x data set. All training times are provided in seconds.

|  | SchNet | PaiNN | SpinConv | eSCN | Equiformer v2 |
|---|---|---|---|---|---|
| Baseline | 7.51 | 8.02 | 33.46 | 100.71 | 57.79 |
| PIWSL | 12.84 | 23.48 | 86.28 | 328.48 | 177.55 |

Table A11 provides training times measured for experiments with and without PIWSL. The training time is measured for a single training epoch and is averaged over five epochs in total. The experiments were performed using 1000 training configurations from the ANI-1x data set. We used a mini-batch of six. The table indicates that PIWSL increases the training time by a factor of two to three compared to the baseline (due to the additional gradient calculations). This is primarily because our PITC and PISC losses effectively double or triple the number of data labels, resulting in a proportional increase in training time due to the expanded set of training labels. We emphasize that the PIWSL approach only alters training time; the inference time is unaffected.

## F.3 Different Configurations for the Physics-Informed Spatial-Consistency Loss

### F.3.1 Triangle-Based

Table A12: **Results for different configurations of the PISC loss.** The presented numerical values are the root mean square errors (RMSEs) for the ANI-1x data set (Smith et al., 2020). Energy (in kcal/mol) and force (in kcal/mol/Å) errors are obtained by averaging over three independent runs. All models are trained using 1000 configurations. The case 1, 2, and 3 correspond to Eq. (4), Eq. (A8) and Eq. (A9), respectively.

| Model | | Baseline | PISC (Case 1) | PISC (Case 2) | PISC (Case 3) |
|---|---|---|---|---|---|
| PaiNN | E | 60.11 | **45.24** | 46.32 | 57.29 |
| | F | 13.10 | **12.33** | 12.42 | 13.28 |

In section 4.2, we consider the following form of the PISC loss

$$L_{\text{PISC}}\left(\mathcal{S}; \boldsymbol{\theta}\right) = \ell\left(E\left(\mathcal{S}_{\delta\mathbf{r}}; \boldsymbol{\theta}\right), E\left(\mathcal{S}_{\delta\mathbf{r}'}; \boldsymbol{\theta}\right) - \sum_{i=1}^{N_{\text{at}}}\langle\delta\mathbf{r}_i'', \mathbf{F}_i\left(\mathcal{S}_{\delta\mathbf{r}'}; \boldsymbol{\theta}\right)\rangle\right), \quad (\text{A7})$$

where $\delta\mathbf{r}, \delta\mathbf{r}', \delta\mathbf{r}''$ are related as $\delta\mathbf{r}' + \delta\mathbf{r}'' = \delta\mathbf{r}$. In this section, as a variant of Eq. (4), we also consider the following three PISC losses

$$L_{\text{PISC,Case 2}}(\mathcal{S};\boldsymbol{\theta}) = \ell\left(E(\mathcal{S};\boldsymbol{\theta}) - \sum_{i=1}^{N_{\text{at}}}\langle\delta\mathbf{r}_i, \mathbf{F}_i(\mathcal{S};\boldsymbol{\theta})\rangle, E(\mathcal{S}_{\delta\mathbf{r}'};\boldsymbol{\theta}) - \sum_{i=1}^{N_{\text{at}}}\langle\delta\mathbf{r}_i'', \mathbf{F}_i(\mathcal{S}_{\delta\mathbf{r}'};\boldsymbol{\theta})\rangle\right),$$
(A8)

$$L_{\text{PISC,Case 3}}(\mathcal{S};\boldsymbol{\theta}) = \ell\left(E(\mathcal{S}_{\delta\mathbf{r}'};\boldsymbol{\theta}), E(\mathcal{S}_{\delta\mathbf{r}};\boldsymbol{\theta}) - \sum_{i=1}^{N_{\text{at}}}\langle-\delta\mathbf{r}_i'', \mathbf{F}_i(\mathcal{S}_{\delta\mathbf{r}};\boldsymbol{\theta})\rangle\right),$$
(A9)

where the point at $\mathbf{r} + \delta\mathbf{r}$ is the point where PIRC loss is imposed (see Eq. (3)). The results are provided in Table A12 and indicate that Eq. (4) (Case 1) shows a better performance than the other cases for both the potential energy and the force predictions. In this study, we used the ANI-1x data set with 1000 training samples different from the one used to train the model used in the main body to avoid overfitting on the test data set. For the coefficient of the PITC and PISC losses, we used 0.1 and 0.001 with $\epsilon_{\text{max}} = 0.01$.

### F.3.2 Further Variations

The flexibility of the PISC loss allows us to explore additional forms of spatial consistency. For example, instead of using a triangular configuration, we can impose spatial consistency between two points at $\mathbf{r}$ and $\mathbf{r} + \delta\mathbf{r}$, leading to the following expression:

$$L_{\text{PISC,2pt}}(\mathcal{S};\boldsymbol{\theta}) = \ell\left(E(\mathcal{S};\boldsymbol{\theta}), E(\mathcal{S}_{\delta\mathbf{r}};\boldsymbol{\theta}) - \sum_{i=1}^{N_{\text{at}}}\langle-\delta\mathbf{r}_i, \mathbf{F}_i(\mathcal{S}_{\delta\mathbf{r}};\boldsymbol{\theta})\rangle\right).$$
(A10)

While not thoroughly investigated, we empirically observed that this loss delivers competitive performance when applied with the same coefficient value as the PITC loss.

Another potential direction is enforcing a reduction in the curl of the forces. This can be achieved by leveraging Stokes' theorem: $\int_\Sigma \nabla \times \mathbf{F} \cdot d\mathbf{S} = \oint_{\partial\Sigma} \mathbf{F} \cdot d\mathbf{l} = 0$ where $\Sigma$ represents a specific surface regime, and $\partial\Sigma$ denotes its boundary. Similar to the PISC loss, the right-hand side of this equation can be effectively described by considering a triangular configuration, where the midpoints of the three sides correspond to $\mathbf{r}, \mathbf{r} + \delta\mathbf{r}, \mathbf{r} + \delta\mathbf{r}'$.

### F.4 Detailed Setups for Qualitative Analysis

### F.4.1 C–H Potential Energy Profile of Aspirin

Table A13: **Performance of PaiNN employed in figure 2 (c, d).** All the models other than the reference model ($N_{\text{train}} = 1000$) use the OC20's hyper-parameters. For the reference model, we tuned the hyper-paramters of PaiNN model following the original paper (Schütt et al., 2021).

| | | $N_{\text{train}} = 100$ | | $N_{\text{train}} = 200$ | | $N_{\text{train}} = 1000$ |
| | | Baseline | PIWSL | Baseline | PIWSL | Baseline |
|---|---|---|---|---|---|---|
| PaiNN | E | 6.55 | **5.64** | 5.11 | **4.48** | 0.68 |
| | F | 7.38 | 7.36 | 3.95 | 3.97 | 1.44 |

This section describes the detailed setup and procedure for section 5.3. First, we trained PaiNN with and without PIWSL losses using the aspirin data from rMD17 with training set sizes of 100 and 200. For PIWSL, we used $(C_{\text{PITC}}, C_{\text{PISC}}, \epsilon_{\text{max}}) = (1.2, 0.01, 0.015)$. The other experimental setups are the same as for rMD17 experiments presented in section D.1. We used the PaiNN model with gradient-based forces to obtain the reference model and tuned the model hyper-parameter with Optuna (Akiba et al., 2019). The obtained models' performance is provided in Table A13. Then, we prepared the aspirin molecule structures, including the corresponding atomic coordinates and atomic types. For these structures, we perturbed one of the C-H bonds with a bond length from 0.8 Å to 1.8 Å. We prepared 100 structures and estimated the corresponding potential energy with the pre-trained models. The aspirin data is provided in our publicly available source code.

### F.4.2 MD Simulation Stability Analysis

**NVE-Ensembles** This section describes the detailed setup and the procedure for our analysis of MD simulations in section 5.3. Because our implementation builds upon the source code provided by Fu et al. (2023), we used their scripts for performing MD simulations. However, we added a minor modification to enable MD simulations in the microcanonical (NVE) statistical ensemble, i.e., the particle position and velocity are updated with velocity Verlet algorithm(Verlet, 1967) [4]. We set the initial temperature to 300 K and the integration time step to 0.5 fs for all simulations. As defined by Fu et al. (2023), the stability of an MD simulation for a target molecule is defined as the time $T$ during which the bond lengths satisfy the following condition

$$\max_{(i,j) \in \mathcal{B}} |(||\mathbf{x}_i(T) - \mathbf{x}_j(T)||) - b_{i,j}| > \Delta, \tag{A11}$$

where $\mathcal{B}$ denotes the set of all bonds, $\{i, j\}$ denote the pair of bonded atoms, and $b_{i,j}$ denotes the equilibrium bond length. Following Fu et al. (2023), we set $\Delta = 0.5$Å. This definition indicates when the molecule experiences significant structural changes during the MD simulation.

We trained PaiNN and Equiformer v2 models with and without PIWSL losses using the aspirin data from rMD17. We used training set sizes of 100, 200, 500, and 1000. The corresponding stability values are presented in Table A14. The hyperparameters for the PIWSL loss are provided in Table A15. For the training set size of 1000, the hyperparameters are the same as those in Table A3. The results for the stability of the PaiNN (direct and gradient-based force) and Equiformer v2 models are shown in Table A16. To select the hyperparameters in Table A15, a series of MD simulations with a fixed random number seed for the initial atomic velocities was used. For our final stability results in Table A14, a series of MD simulations with three different random seeds for velocities and the selected hyperparameters was used. As the correlation between stability and energy/force errors can be rather weak (see also Fu et al. (2023)), models with stability improved through PIWSL do not necessarily outperform the baseline models in terms of their accuracy. figure A1 presents the total energy difference during MD simulations reported in figure 3, which is measured by the following equation: $|E(t_\mathrm{f}) - E(t_\mathrm{init})|/E(t_\mathrm{init})$, where $t_\mathrm{f}$ and $t_\mathrm{init}$ denote the final and initial time steps. Here $t_\mathrm{f}$ is defined as half of the time step at which the simulation is deemed unstable, corresponding to the reported time step in Table A14. These results indicate that PIWSL slightly enhances the energy conservation capability of MLIP models in most cases, though the improvement is relatively modest due to the small deviations observed.

**NVT-Ensembles** To investigate the effect of thermostats, we also performed MD simulations in the canonical (NVT) statistical ensemble, where temperature is maintained constant. To keep the temperature constant, we used Nosé-Hoover thermostat (Nosé, 1984; Hoover, 1985). The initial and target temperatures were both set to 300 K for all simulations. The integration time step was set to 0.5 fs and the characteristic parameter $\tau$ for the thermostat is set to 20 fs. The result, shown in figure A2, demonstrate that the thermostat stabilizes the simulations by mitigating the increase in kinetic energy.

### F.5 Metric Dependence of PITC

Table A17 provides the result of the metric dependence of PIWSL. For simplicity, we only consider the PITC loss (the coefficient of the PITC and PISC losses are set as 0.1 and 0). For the ReLU metric, we consider

$$L_{\mathrm{ReLU}}(\mathcal{S}; \boldsymbol{\theta}) = \mathrm{ReLU}\left(\left|E(\mathcal{S}; \boldsymbol{\theta}) - \sum_{i=1}^{N_{\mathrm{at}}} \langle \delta \mathbf{r}_i, \mathbf{F}_i(\mathcal{S}; \boldsymbol{\theta})\rangle - E(\mathcal{S}_{\delta \mathbf{r}}; \boldsymbol{\theta})\right| - E(\mathcal{S}_{\delta \mathbf{r}}; \boldsymbol{\theta})||\delta \mathbf{r}||^2\right). \tag{A12}$$

This metric is zero when the difference between the two terms is less than the second-order term in $\delta \mathbf{r}$. The results indicate that taking the second-order term into account does not improve the performance (see PITC MAE Loss and PITC ReLU Loss results), and the MSE loss function shows

---

[4]Note that the total energy conservation necessary for the microcanonical statistical ensemble is in general not perfectly satisfied due to the numerical error, in particular, when the force is not calculated as the curl of the force.

Table A14: **Stability of the models employed in the MD analysis.** The presented numerical values are the stability defined by Eq. (A11) measured in ps. The results are obtained as an average over three different random seeds for the initial velocity of the atoms in the target aspirin molecule. "GF" denotes the gradient-based force prediction.

|  |  | $N_{\text{train}} = 100$ | $N_{\text{train}} = 200$ | $N_{\text{train}} = 500$ | $N_{\text{train}} = 1000$ |
|---|---|---|---|---|---|
| PaiNN | Baseline | $0.60 \pm 0.00$ | $1.35 \pm 0.00$ | $1.45 \pm 0.00$ | $1.70 \pm 0.10$ |
|  | PIWSL | $\mathbf{0.70 \pm 0.00}$ | $\mathbf{1.43 \pm 0.03}$ | $\mathbf{2.55 \pm 0.52}$ | $\mathbf{2.35 \pm 0.22}$ |
| Equiformer | Baseline | $1.05 \pm 0.00$ | $1.45 \pm 0.00$ | $2.28 \pm 0.49$ | $4.18 \pm 0.33$ |
|  | PIWSL | $\mathbf{1.17 \pm 0.10}$ | $\mathbf{2.43 \pm 0.06}$ | $\mathbf{3.20 \pm 0.33}$ | $\mathbf{5.77 \pm 1.84}$ |
| PaiNN-GF | Baseline | $\mathbf{3.25 \pm 3.98}$ | $\mathbf{220.5 \pm 137.7}$ | – | – |
|  | PIWSL | $\mathbf{15.07 \pm 10.09}$ | $\mathbf{267.7 \pm 56.0}$ | – | – |

Table A15: **Hyper-parameters for the PIWSL loss used in the MD analysis.** We used the following hyper-parameter for MD simulation analysis: $(C_{\text{PITC}}, C_{\text{PISC}}, \epsilon_{\max})=$ Case $\alpha$: (0.01, 0.001, 0.025), Case $\beta$: (1.2, 0.01, 0.01), Case $\gamma$: (1.2, 0.01, 0.025), Case $\delta$: (1.2, 0.01, 0.015), Case $\epsilon$: (0.1, 0.01, 0.01), and Case $\zeta$: (1.0, 0., 0.01). "GF" denotes the gradient-based force prediction.

| Dataset | Size | Equiformer v2 | PaiNN | PaiNN-GF |
|---|---|---|---|---|
| rMD17 | 100 | $\alpha$ | $\delta$ | $\epsilon$ |
| (Aspirin) | 200 | $\beta$ | $\beta$ | $\zeta$ |
|  | 500 | $\gamma$ | $\gamma$ | – |

the best performance. In this study, we used the ANI-1x data set and the 1000 training samples. These samples differ from the one used to train the model in the main text to avoid overfitting the test data set.

### F.6 PERTURBATION MAGNITUDE DEPENDENCE OF PITC

In this section, we provide the result of the perturbation magnitude dependence of PIWSL, i.e., $\|\delta \mathbf{r}\| = \epsilon$. For simplicity, we only consider the PITC loss (the coefficient of the PITC and PISC losses are set as 0.1 and 0.0). The results are provided in Table A18 and demonstrate that the longer perturbation vector length is fruitful for force predictions. However, values that are too large are harmful to predicting potential energy. In this study, we used the ANI-1x data set and the 1000 training samples. These samples differ from the one used to train the model in the main text to avoid overfitting the test data set.

### F.7 DEPENDENCE OF PITC ON THE NUMBER OF PERTURBED ATOMS

This section provides the result of the perturbed atom number dependence of PIWSL. For simplicity, we only consider the PITC loss (the coefficient of the PIRC and PISC losses are set as 0.1 and 0). In this study, we randomly selected atoms in a training sample following the ratio of 0 %, 10 %, 20 %, 50 %, 75 %, 90 %, 100 %. The results are provided in Table A19, which indicates that around 75% to 100% ratio cases result in the best performance for the force and the potential energy prediction. However, the number dependence is rather complicated. Therefore, in the main text, we perturbed all the atoms (100 %) as a conservative choice. In this study, we used the ANI-1x data set and the 1000 training samples. These samples differ from the one used to train the model in the main text to avoid overfitting the test data set.

### F.8 DEPENDENCE ON THE NUMBER OF TRAINING ITERATIONS

To show the effectiveness of our approach even in the case of longer training, we provide the result of the dependence of PIWSL on the number of training iterations. In this study, we performed training twice as long as in the main text, that is, 80,000 iterations for ANI-1x with 1000 training samples.

Table A16: **Energy and force errors for the models employed in the MD analysis.** The presented numerical values are the root-mean-square errors (RMSEs) of energy (E) and force (F). Energy RMSE is given in kcal/mol, while force RMSE is in kcal/mol/Å. "GF" denotes the gradient-based force prediction.

| | | $N_{\text{train}} = 100$ | | $N_{\text{train}} = 200$ | | $N_{\text{train}} = 500$ | |
| | | Baseline | PIWSL | Baseline | PIWSL | Baseline | PIWSL |
|---|---|---|---|---|---|---|---|
| PaiNN-DF | E | 6.55 | 6.49 | 5.11 | 4.56 | 4.50 | 4.64 |
| | F | 7.38 | 7.37 | 3.95 | 3.99 | 2.55 | 2.51 |
| Equiformer v2 | E | 4.79 | 4.64 | 4.92 | 4.82 | 3.37 | 3.77 |
| | F | 4.86 | 4.90 | 2.50 | 2.42 | 1.91 | 1.91 |
| PaiNN-GF | E | 6.05 | 6.03 | 6.01 | 6.02 | – | – |
| | F | 6.41 | 6.33 | 3.50 | 3.53 | – | – |

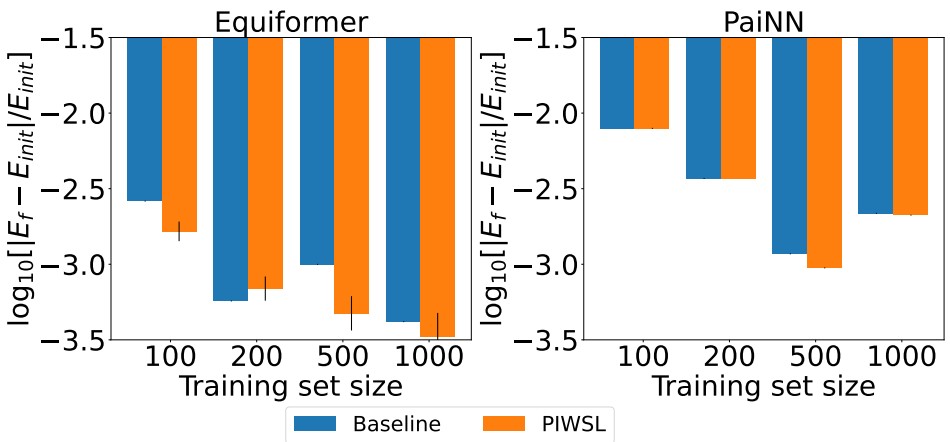

Figure A1: **Analysis of the total energy conservation uring MD simulations with MLIP models.** The amount of the change of the total energy during MD simulations is assessed for the baseline MLIP models and those trained with PIWSL. The total energy is measured at the inital and final time-step and the difference is normalized by the total energy at the initial time-step. All results are obtained for the aspirin molecule and MD simulations in the microcanonical (N V E) statistical ensemble.

The results are provided in Table A20 and indicate that our approach performs better in the longer training case. On the other hand, the training without PIWSL shows an overfitting to the validation data set, reducing its performance compared to the shorter training case. In this study, we used the ANI-1x data set and the 1000 training samples. These samples differ from the one used to train the model in the main text to avoid overfitting the test data set. The coefficients of the PITC and PISC losses are $1.2$ and $0.01$, respectively.

### F.9 ADDITIONAL EXPERIMENTS WITH GRADIENT-BASED FORCES

In this section, we provide the result of the training with the gradient-based force predictions. The results are provided in Table A21 and demonstrate that our PIWSL loss enables a better force prediction, even in the case of gradient-based force predictions. These results also indicate that our PIWSL method can improve the ML model performance in the case of MLIPs commonly applied in computational chemistry and materials science. We consider that this is partly due to the effectiveness of the weak label at $\mathbf{r} + \delta\mathbf{r}$ as indicated by the WL results, which show an improvement of the performance different from the case with the direct force branch (see also section 5.5). We hypothesize that the further improvement results from the additional gradient calculation as indicated

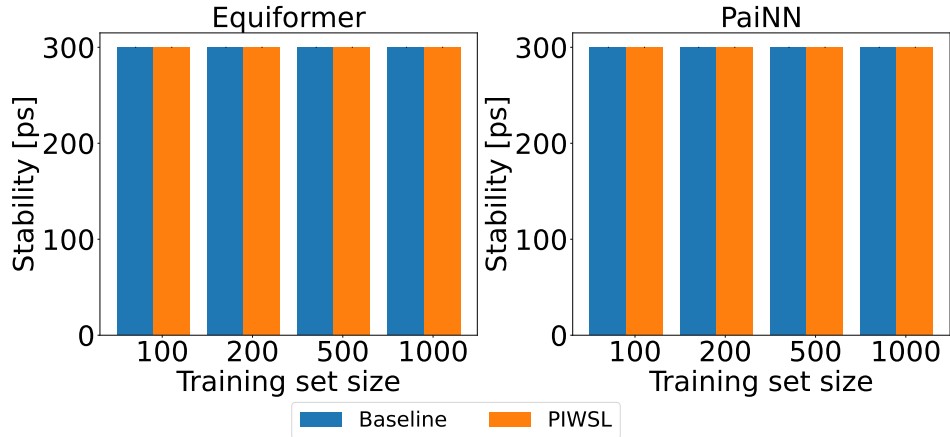

Figure A2: **Stability analysis of the MLIP models during MD simulations.** Stability during MD simulations is assessed for the baseline MLIP models and those trained with PIWSL. All results are obtained for the aspirin molecule and MD simulations in the canonical (N V T) statistical ensemble. We measure stability during MD simulations according to (Fu et al., 2023).

Table A17: **Metric dependence of PITC.** The presented numerical values are the root mean square errors (RMSEs) for the ANI-1x data set (Smith et al., 2020). Energy (in kcal/mol) and force (in kcal/mol/Å) errors are obtained by averaging over three independent runs. All models are trained using 1000 configurations. MAE refers to the mean absolute error, and MSE denotes the mean square error.

| Model | | Baseline | PITC MAE Loss | PITC MSE Loss | PITC ReLU Loss |
|---|---|---|---|---|---|
| PaiNN | E | 60.11 | 58.84 | **47.09** | 60.47 |
| | F | 13.10 | 13.18 | **12.19** | 13.06 |

in Eq. (A5) and Eq. (A6). This observation also indicates that our PIWSL method can potentially improve other generic property prediction tasks by calculating their first derivatives in terms of the atomic coordinate and utilizing the proposed loss functions. In this study, the coefficient of the PITC and PISC losses are set as $0.1$ and $0.01$ with $\epsilon_{\max} = 0.01$. The weak label loss coefficient is set as $0.1$.

### F.10 REDUCING CURL OF FORCES FOR MODELS WITH THE FORCE BRANCH

In this section, we study the effect of our loss functions on the curl of forces in the case of the model with the force branch. The results are provided in Table A22, which shows that our PITC loss reduces the curl of the predicted forces, allowing potentially better energy conservation during MD simulations. In this study, we used the ANI-1x data set and the 1000 training samples. These samples differ from the one used to train the model in the main text to avoid overfitting the test data set. The hyper-parameters of the PITC and PISC losses are $(C_{\mathrm{PITC}}, C_{\mathrm{PISC}}, \epsilon_{\max}) = (1.2, 0.01, 0.025)$. It is theoretically possible to define a loss function aimed at directly minimizing the absolute value of the curl of forces. However, this approach necessitates calculating the Hessian matrix, which requires a substantial memory cost given the limitations of current computational resources. Developing a method to train with such a loss function while mitigating memory requirements is a promising direction for future research.

Table A18: **Perturbation magnitude dependence of PITC.** The presented numerical values are the root mean square errors (RMSEs) for the ANI-1x data set (Smith et al., 2020). Energy (in kcal/mol) and force (in kcal/mol/Å) errors are obtained by averaging over three independent runs. All models are trained using 1000 configurations.

| Model | | Baseline | $\epsilon_{\max} = 0.0005$ | $\epsilon_{\max} = 0.005$ | $\epsilon_{\max} = 0.05$ |
|---|---|---|---|---|---|
| PaiNN | E | 60.11 | 60.43 | **47.09** | 109.17 |
| | F | 13.10 | 12.75 | 12.19 | **11.70** |

Table A19: **Dependence of PITC on the number of perturbed atoms.** The presented numerical values are the root mean square errors (RMSEs) for the ANI-1x data set (Smith et al., 2020). Energy (in kcal/mol) and force (in kcal/mol/Å) errors are obtained by averaging over three independent runs. All models are trained using 1000 configurations.

| Model | | Baseline | 10% | 20% | 50% | 75% | 90% | 100 % |
|---|---|---|---|---|---|---|---|---|
| PaiNN | E | 60.11 | 46.68 | 52.37 | 54.51 | 46.94 | **45.92** | 46.32 |
| | F | 13.10 | 13.03 | 12.62 | 12.16 | **12.14** | 12.24 | 12.42 |

Table A20: **Dependence on the number of training iterations.** The presented numerical values are the root mean square errors (RMSEs) for the ANI-1x data set (Smith et al., 2020). Energy (in kcal/mol) and force (in kcal/mol/Å) errors are obtained by averaging over three independent runs. All models are trained using 1000 configurations.

| Model | Iteration Number | | Baseline | PIWSL |
|---|---|---|---|---|
| PaiNN | 40,000 | E | $56.62 \pm 2.80$ | $\mathbf{24.53 \pm 0.16}$ |
| | | F | $12.96 \pm 0.18$ | $\mathbf{11.43 \pm 0.05}$ |
| | 80,000 | E | $59.92 \pm 1.47$ | $\mathbf{23.78 \pm 0.16}$ |
| | | F | $13.10 \pm 0.19$ | $\mathbf{11.50 \pm 0.04}$ |

Table A21: **Results of PIWSL with gradient-based force predictions.** The presented numerical values are the root mean square errors (RMSEs) for the ANI-1x data set (Smith et al., 2020). Energy (in kcal/mol) and force (in kcal/mol/Å) errors are obtained by averaging over three independent runs. All models are trained using 1000 configurations.

| Model | | Baseline (GF) | PIWSL (GF) | WL (GF) |
|---|---|---|---|---|
| PaiNN | E | $23.57 \pm 0.62$ | $\mathbf{20.23 \pm 0.18}$ | $22.61 \pm 0.50$ |
| | F | $11.32 \pm 0.08$ | $\mathbf{11.13 \pm 0.04}$ | $11.72 \pm 0.06$ |
| Equiformer | E | $29.07 \pm 2.32$ | $\mathbf{19.53 \pm 0.32}$ | $21.07 \pm 0.86$ |
| | F | $\mathbf{11.90 \pm 0.13}$ | $11.99 \pm 0.03$ | $\mathbf{11.90 \pm 0.20}$ |

Table A22: **Curl of forces for models with the force branch.** The presented numerical values are the absolute values of the total curl of the force evaluated for the ANI-1x data set (Smith et al., 2020). Energy (in kcal/mol) and force (in kcal/mol/Å) errors are obtained by averaging over three independent runs. All models are trained using 1000 configurations.

| Model | Baseline | PITC |
|---|---|---|
| PaiNN | $45.18 \pm 4.07$ | $\mathbf{39.06 \pm 0.58}$ |
| Equiformer | $29.62 \pm 0.28$ | $\mathbf{23.42 \pm 0.09}$ |

