# OpenReview forum: "Physics-Informed Weakly Supervised Learning for Interatomic Potentials"
_ICLR.cc/2025/Conference — Submitted to ICLR 2025_

### Official Review · Reviewer_xnGz · 2024-10-22

**Soundness:** 4
**Presentation:** 3
**Contribution:** 3
**Rating:** 6
**Confidence:** 5

**Summary:**

This work presents a physics-informed, weakly supervised approach for training machine-learned interatomic potentials. The method is innovative, and the ideas are inspiring. However, it may not be practical for real-world applications, highlighting the need for further validation and case studies. Additionally, the improvements in experimental results are minimal, particularly with the rMD17 dataset, a common benchmark.

**Strengths:**

1. The work introduces a physics-informed, weakly supervised approach for training machine-learned interatomic potentials.
2. The method is innovative, and the ideas are truly inspiring.

**Weaknesses:**

The proposed method may not be practical for real-world applications, highlighting the need for further validation and case studies. Additionally, the improvement in experimental results is minimal, particularly with the rMD17 dataset, which is a common benchmark.

**Questions:**

In training MLIPs for small molecules, there is usually a substantial amount of data available (over 1,000 samples). How effective is the proposed method with a larger dataset of small molecules?

How effective is training on datasets for large molecules, such as the MD22 dataset?

---

> ### Author Response · Authors · 2024-11-22
> **reply**
>
> $\textbf{[W1] [Q1] Training on small molecule data set}$
>
> > The proposed method may not be practical for real-world applications, highlighting the need for further validation and case studies.
>
> Following your suggestion, as outlined in the “General Remark,” we conducted several new experiments, including those with 1,000,000 and 5,000,000 samples from the ANI-1x dataset, MD22 with MACE, and various MD simulations using MLIP models.
>
> > Additionally, the improvement in experimental results is minimal,
>
> We would like to respectfully point out that PIWSL leads to a significant reduction in error for the ANI-1x and TiO2 datasets across several models. While the error remains non-negligible due to the sparse data regime, the error reduction in such regimes is crucial for real-world applications. This is because, in these scenarios, new data and labels must be generated using computationally expensive DFT or coupled-cluster methods. By reducing errors, the MLIP model enables longer MD simulations, reducing the overall time and cost for data generation and model training. Please refer to the “General Remark” for further details.
>
> > particularly with the rMD17 dataset, which is a common benchmark.
>
> In the case of MD17, the dataset is homogeneous (containing conformations of a single molecule), and 1,000 samples are sufficient to cover the possible variations, making PIWSL less necessary. This is supported by the MD17 dataset authors, who note on their homepage (https://figshare.com/articles/dataset/Revised_MD17_dataset_rMD17_/12672038)
> ) that:
>
> >One warning: As the structures are taken from a molecular dynamics simulation (i.e. time series data), they are not guaranteed to be independent samples. This is easily evident from the autocorrelation function for the original MD17 dataset. In short: DO NOT train a model on more than 1000 samples from this dataset. Data already published with 50K samples on the original MD17 dataset should be considered meaningless due to this fact and due to the noise in the original data.
>
> On the other hand, the heterogeneous ANI-1x dataset (which includes various molecules and uses diverse sampling methods) shows significant improvement with PIWSL.
>
> $\textbf{[Q2] Training on MD22 data set}$
>
> As mentioned in the “General Remark,” we added a new experiment using MACE on the MD22 dataset. This demonstrates that PIWSL remains effective for large molecules, even when applied to a state-of-the-art model. While the improvement on 600 samples is not substantial, PIWSL provides a notable enhancement in potential energy, particularly in the sparse data regime, which is the primary focus of our work.

---

> > ### Comment · Reviewer_xnGz · 2024-12-03
> >
> > Thank you for your reply. I am glad that you have added the results of training on a larger dataset. Based on your latest experimental results and the findings in your paper, I will further consider whether to improve my score

---

> ### Author Response · Authors · 2024-11-29
>
> Dear Reviewer xnGz,
>
> Thank you very much for your thoughtful feedback and interest in our work. We have improved and validate the effectiveness of PIWSL further by incorporating the following major updates: (1) demonstrating the effectiveness of PIWSL in the context of the foundation model + fine-tuning scenario, and (2) introducing a second-order extension of PITC (both are described in the general remark. )
> We would greatly appreciate any additional feedback or discussion points that could further improve our work and help elevate your present evaluation.
>
> Best regards,
>
> The Authors

---

> > ### Author Response · Authors · 2024-12-02
> > **Any last minuets discussion points?**
> >
> > Dear Reviewe xnGz,
> >
> > As the discussion period approaches its conclusion within the next 24 hours, we would greatly appreciate it if you could let us know of any remaining discussion points that you feel require clarification or further elaboration to support a potential adjustment to your rating.
> >
> > Thank you once again for your valuable feedback and assistance throughout this process.
> >
> > Best regards,
> >
> > Authors

---

### Official Review · Reviewer_ji8n · 2024-10-27

**Soundness:** 3
**Presentation:** 3
**Contribution:** 3
**Rating:** 8
**Confidence:** 3

**Summary:**

The paper proposes an extended loss for the training of machine learning interatomic potentials. Two additional terms are added in the loss, the physics-informed Taylor-expansion-based consistency (PITC) loss and the physics-informed spatial-consistency (PISC) loss. The idea behind these terms is to generate additional approximate training data points by a first-order Taylor expansion of the energy at a given training data point. The goal is to increase the robustness of the machine learning interatomic potential for applications in this training approach. The approach is validated in several benchmarks.

**Strengths:**

- The approach appears to provide consistently better results.
- The approach is independent on the machine learning potential model.
- The approach can be easily integrated in other projects.

**Weaknesses:**

- The potential energy curve of the C-H bond is badly represented in both cases, with and without PIWSL. For me, the argument is not strong if it still fails but not as badly. Maybe this has just to be trained to better/more (400, 800) training data and then both will give good results. To show the advantage of PIWSL, I think you have to find the sweet spot where PIWSL can already deal with a low amount of data and the conventional approach fails due to the lack of data. You could then also add a graph, where both approaches are trained on more data and both succeed to demonstrate that it is really the more efficient data handling of PIWSL. I think, the reference and predicted curves should at least agree in the range where the relative energy is below about 2 kcal/mol, because this region will be sampled in molecular dynamics simulations. However, especially the error for bond increases in plot d appears very large (>1 kcal/mol), currently.
- The approach increases the computational time required for training significantly.

**Questions:**

- Are the reported E and F RMSE values for training or test data?
- If I understand it correctly, the training was stopped after a per-defined number of training epochs. However, the PIWSL method requires 2-3 times more training time due to the additional calculation. I think a fair comparison would be to reduce the number of training epochs for PIWSL by a factor of 2-3 or otherwise increase the number of training epochs for the baseline approach. You could also just use the wall-clock time as stopping criterion instead of a given number of epochs. Did you check the convergence of the baseline approach? You could add learning curves to validate the convergence.

---

> ### Author Response · Authors · 2024-11-22
> **reply**
>
> $\textbf{[W1] Figure 2-c,d}$
>
> Thank you very much for your thoughtful comment. The bond lengths for aspirin configurations in the MD17 dataset mainly range from 1.05 Å to 1.15 Å, (with a Gaussian distribution centered around 1.1 Å and a standard deviation of approximately 0.05 Å). Therefore, it is not feasible for the ML models to fit over the range of [1.05, 1.15] Å, as shown in Figures 2(c,d). From this perspective, we believe the current improvement is still meaningful. One possible solution would be to reconsider the plot range to cover 1.0 to 1.2 Å. We welcome any further suggestions to improve the draft.
>
> $\textbf{[W2] Increase of training time}$
>
> We would like to respectfully clarify that the increase in training time (2-3x) arises simply from the increased number of training samples (weak labels) introduced by PITC and PISC, which effectively scales the number of labels by 2-3 times. Therefore, this increase in training time is a natural consequence, simply scaling with the number of labels. Based on this observation, one possible solution would be to reduce the number of perturbed atoms, as demonstrated in Section F.6 and Table A.19, where perturbing 50–75% of the atoms still results in a competitive performance gain.
>
> Additionally, the increase in training time is negligible compared to the substantial computational cost and time required to generate data, especially when using DFT or coupled-cluster methods. This is particularly true as the molecule size increases, where the computational resources required for data generation grow significantly.
>
> $\textbf{[Q1]  Are the reported E and F RMSE values for training or test data?}$
>
> The reported results are on Test data and now we explicitly mentioned this in the new draft.
>
> $\textbf{[Q2] Convergence check}$
>
> Thank you for your thoughtful comment. Defining a "fair comparison" can be challenging. In this work, we take a practical perspective: training with a fixed number of iterations is often more beneficial, as it is typically faster than generating new data, as discussed in W2.
>
> > Did you check the convergence of the baseline approach?
>
> Table A19 provides a comparison with longer training iterations, which shows that while the baseline method overfits the training data, PIWSL still improves its performance.

---

### Official Review · Reviewer_xuZ7 · 2024-10-27

**Soundness:** 2
**Presentation:** 4
**Contribution:** 2
**Rating:** 3
**Confidence:** 5

**Summary:**

The authors propose a new pre-training method for MLIPs by augmenting the typical supervised loss with a physics-inspired loss (PIWSL). The proposed objective has a part that encourages the model to learn a conservative force field, and another part that ensures consistency of the potential energy surface based on a Taylor expansion around structure in the dataset. The authors conduct experiments on small datasets showing that their physics-informed loss improves performance compared to training just with supervision on energies and forces.

**Strengths:**

Strengths:
- The paper is written clearly and the method is motivated and presented well.
- Pre-training / self-supervised methods are important for the field of ML and could be particularly impactful for the field of MLIPs, where data is computationally expensive to generate.
- The proposed method is novel and physically motivated. It seems to be easy to implement programmatically and could be applied generally to any MLIP.

**Weaknesses:**

The main weaknesses are in the evaluation of the proposed method. In most other areas of ML, weakly-supervised learning is done with large datasets to learn general representations that can then be transferred / fine tuned downstream. However, in this paper, the experiments are done on extremely small datasets ranging in size from 100 - 1000 samples. This makes it hard to argue that the PIWSL serves as a useful objective, especially as the field migrates to larger datasets (OC20, MPTrj, SPICE, etc.). Concretely (roughly in order of importance):
- The experiments are done on extremely small datasets, making it hard to argue that the PIWSL helps the model learn useful representations (I suspect that any form of regularization would improve performance on datasets this small). On these small datasets, a non-deep learning based model like sGDML [3] outperforms all of the models tested in this paper in Table A9 by a large margin. It would be much more compelling if PIWSL still improved performance on the larger (1M+ samples) more common datasets (OC20, SPICE, MPTrj, full ANI-1x dataset, etc.).
- Since the models are trained on such small datasets, the errors are currently huge (more than 10 kcal / mol across in almost all cases in table 1 for instance). Therefore, the improvement of ~1 kcal / mol with PIWSL still leaves the models with a huge error (reflected in the poor simulation performance in Figure 3.) As the dataset size increases, the improvement from PIWSL seems to decrease as well (mentioned by the authors in Fig. 2).
- The baseline NoisyNodes isn’t really a valid comparison since it is meant to only work for equilibrium structures. A much more valid comparison would be something like DeNS [1] that does work for the non-equilibrium structures that the authors test on.
- The models tested do not necessarily cover the state-of-the-art models for these types of molecular datasets. Equiformer and eSCN for instance are tuned for OC20 and are often outperformed by models like MACE [2] on molecular datasets. Something like MACE is also often able to get errors well below 1 kcal / mol with just supervision from energies and forces. If PIWSL still helped with these better models, that would be more compelling.

Weakly/self-supervised strategies have shown to be extremely effective in other fields of ML, and I am optimistic that methods like the ones presented by the authors will be important for MLIPs. However, the current evaluations in the paper do not provide convincing evidence that the current instantiation of PIWSL would actually help the best MLIPs when trained on large datasets. I would be happy to raise my score with more rigorous evaluations.

**Questions:**

- Why were the simulations done with the NVE ensemble (it makes it harder to compare to the work in Fu et. al. (2023) since they do an NVT with NoseHoover)? This is briefly discussed around line 400 but I feel like some results with NVT simulations could be helpful, since at the moment the NVE simulations yield very low stability.
- What happens if the models are trained explicitly to minimize curl as opposed to the PISC loss?

References:
[1] Yi-Lun Liao, Tess Smidt, Muhammed Shuaibi, & Abhishek Das. (2024). Generalizing Denoising to Non-Equilibrium Structures Improves Equivariant Force Fields.
[2] Ilyes Batatia, Dávid Péter Kovács, Gregor N. C. Simm, Christoph Ortner, & Gábor Csányi. (2023). MACE: Higher Order Equivariant Message Passing Neural Networks for Fast and Accurate Force Fields
[3] Stefan Chmiela, Valentin Vassilev-Galindo, Oliver T. Unke, Adil Kabylda, Huziel E. Sauceda, Alexandre Tkatchenko, & Klaus-Robert Müller (2023). Accurate global machine learning force fields for molecules with hundreds of atoms. Science Advances, 9(2), eadf0873.

---

> ### Author Response · Authors · 2024-11-22
> **reply**
>
> $\textbf{[W1] Data set size}$
>
> Following the reviewer’s suggestion, we conducted additional experiments using 1,000,000 and 5,000,000 samples from the ANI-1x dataset. The results, provided in Table A5 of the revised manuscript, demonstrate that PIWSL improves the energy prediction error by approximately 5–10%. For further details, please refer to the “General Remark.”
>
> >I suspect that any form of regularization would improve performance on datasets this small)
>
> This is an interesting observation. The effectiveness of regularization depends significantly on the number of samples per molecule and the sampling method. For example, in the case of MD17, which is a homogeneous dataset containing configurations of a single molecule, 1,000 samples may adequately cover the range of possible variations, potentially reducing the need for PIWSL. This notion is partly supported by the creators of MD17, who state on their homepage (https://figshare.com/articles/dataset/Revised_MD17_dataset_rMD17_/12672038)
> ) as:
>
> >One warning: As the structures are taken from a molecular dynamics simulation (i.e. time series data), they are not guaranteed to be independent samples. This is easily evident from the autocorrelation function for the original MD17 dataset. In short: DO NOT train a model on more than 1000 samples from this dataset. Data already published with 50K samples on the original MD17 dataset should be considered meaningless due to this fact and due to the noise in the original data.
>
> In contrast, the ANI-1x dataset, which is heterogeneous (containing various molecules) and sampled using diverse methods, demonstrates significant improvement when using PIWSL.
>
> $\textbf{[W2] Not small error due to small data set size}$
>
> As mentioned in the “General Remark,” we would like to emphasize that the primary focus of our work is the sparse data regime, a common scenario in real-world computational chemistry applications. In such cases, training data for MLIP models must often be generated from scratch using DFT solvers. Reducing errors in sparse data regimes—while it might be perceived as making a "bad" ML model less "bad"—is crucial because it minimizes the number of required DFT calculations, extends the duration of atomistic simulations that can be performed with the surrogate model, and reduces the overall training cost for developing a reliable MLIP model. Please also note that even in the large data regime, more than 1,000,000 samples, the performance gain from PIWSL remains around 5–10 %, not becoming negligible.
>
> > the errors are currently huge (more than 10 kcal / mol across in almost all cases in table 1 for instance)
>
> This observation is partly explained by using the root mean square error (RMSE) as the evaluation metric, compared to the mean absolute error (MAE), which is more commonly used in this community. RMSE values are typically more than two times larger than MAE values. Our motivation for using RMSE is to capture the effect of outliers, which is crucial for MD simulations with MLIP models. Outliers can induce unstable behavior during simulations, making RMSE a more appropriate metric for assessing stability.
>
> $\textbf{[W3] Other possible baseline}$
>
> Thank you very much for bringing this interesting work to our attention. As far as we know, DeNS is still in draft form (it only appeared on arXiv), and the authors may release an official version once a future conference accepts the paper. Therefore, we will carefully monitor any updates and have cited it in our paper as related work (Section D.2).
>
> $\textbf{[W4] MACE model}$
>
> Following the kind advice, we tried the MACE model on MD22 and observed an improvement by PIWSL (Table A10). Please also refer to “General Remark”.

---

> ### Author Response · Authors · 2024-11-22
> **reply**
>
> $\textbf{[W-gen] Primary target of our physics-informed weakly supervised approach}$
>
> We would like to respectfully clarify that our weakly supervised strategy is not intended to assign pseudo-labels to large amounts of unlabeled data. This distinction arises because, unlike in other application domains such as image processing or NLP, no large pool of unlabeled data is available for MLIP models. Moreover, generating a minimal yet sufficient set of configurations is non-trivial (see Appendix C). Our primary focus is the sparse data regime, where training data for MLIP models must be generated from scratch using DFT solvers, as highlighted in the introduction, abstract, and conclusion. For further clarification, please refer to the “General Remark.”
>
> > However, the current evaluations in the paper do not provide convincing evidence that the current instantiation of PIWSL would actually help the best MLIPs when trained on large datasets.
>
> We hope our new results for 1,000,000 and 5,000,000 samples from the ANI-1x dataset and the MACE model trained on MD22 address the reviewer’s concern. Please let us know if that is not the case.
>
> $\textbf{[Q1] NVT simulation}$
>
> Following the reviewer’s suggestion, we have added simulation results using NVT statistical ensembles in Figure A2. These results show that the thermostat significantly stabilizes the simulation, effectively masking the true stability characteristics of the MLIP models.
>
> $\textbf{[Q2] directly reducing curl of force}$
>
> Thank you very much for this insightful proposal! While it is theoretically possible to define a loss function aimed at directly minimizing the absolute value of the curl of forces, this approach requires calculating the Hessian, which involves significant memory costs given the current computational limitations. Therefore, we believe it is necessary to develop a method for training with such a loss function while mitigating memory requirements (see also Section F.10). Another promising direction would be combining the PISC loss with Stokes’ theorem, which could shift the focus from reducing the curl of the force prediction to minimizing the integrated force along the line (please also refer to Section F3.2). Although we have not yet explored this approach, it highlights the flexibility of the PIWSL method, offering various potential applications beyond the formalism we proposed.

---

### Official Review · Reviewer_kYHe · 2024-11-09

**Soundness:** 2
**Presentation:** 2
**Contribution:** 2
**Rating:** 3
**Confidence:** 4

**Summary:**

The authors propose two self-supervised losses to train machine learning interatomic potentials. One is based on a Taylor-expansion approach, and the other is based on a consistency loss. They use these losses to train MLIPs on ANI, a TiO2 dataset, and some molecules in rMD17. They look at training models with mostly n=100 to n=1000 samples, and performance primarily on energy and forces prediction, as well as an example of running the molecular dynamics simulation.

**Strengths:**

- Self-supervised learning strategies for MLIPs are not very common, so it is nice to see some attempts towards developing these.
- Using these losses could be useful in cases where only energy labels are available, and not forces

**Weaknesses:**

There are many concerns I have with this method and actual utility in the broader scope of the MLIP literature.

- The results are not very convincing that this is a method that would actually get used. The authors are looking at a low number of data points, when larger datasets now exist. It is true that they are looking at more data points in the ANI dataset, though the level of accuracy and data generation of this dataset is suspect.

- In some cases, models can be trained by taking gradients of the energy to get a conservative force field, including cases the authors looked at. If this is the case, why do we need these losses? How do these losses compare in accuracy to taking gradients of the energy? This would make sense to do in datasets like rMD17 and other molecular datasets that have the forces.

- Stability analysis is only done for 3-6 ps which is very, very short. The models are also barely improving and still not usable, and it is not a good sign that it does not do well for such a simple molecule like aspirin. Why are the authors doing NVE simulations instead of NVT? Why not try NVT?

- The poor molecular dynamics performance means that these potentials are not very good, so they cannot be used for actual simulations.

- The authors say that they run NVE simulations and that this means the total energy is conserved. This is not necessarily true, because the authors may not have a potential that actually conserves energy. The authors can test actual energy conservation with their model.

- There is a line on 361 about “this observation indicates strong evidence of the effectiveness of PIWSL applied to bulk materials.” This is a very specific dataset (TiO2) and such claims are much greater than what is actually being presented. If the authors want to go towards such claims, they should be looking at datasets like the Materials Project, which is much more representative of bulk materials.

- Besides the above example, there are too many strong conclusions are being drawn from not great results to back these up, as these experiments are not being done on more representative datasets and sizes.

**Questions:**

- Do these loss terms help for more representative benchmarked datasets like Materials Project? Or MD22, which is a more representative dataset than MD17

- How does using these losses compare to taking energy gradients when the forces are known?

- Do the losses actually improve energy conservation?

---

> ### Author Response · Authors · 2024-11-22
> **reply**
>
> $\textbf{[W1] Dataset size}$
>
> As mentioned in the general remark to all reviewers, we would like to emphasize that the primary focus of our work is the sparse data regime, a common scenario in real-world computational chemistry applications. Here, training data for MLIP models should be generated from scratch using DFT calculations. We emphasize that a reduction of the error in sparse data regimes – even though it could be considered as making a "bad" ML model less "bad"-- is crucial because it minimizes the number of required DFT calculations, extends the duration of atomistic simulations with the surrogate model, and reduces the overall training cost for developing a reliable MLIP model.
>
> On the other hand, we acknowledge that it is also valuable to assess the performance of PIWSL in the large data regime. To this end, we conducted additional experiments using 1,000,000 and 5,000,000 samples from the ANI-1x data set. The results show that the proposed method improves the accuracy of the PaiNN model in the large data regime, demonstrating the robustness and effectiveness of PIWSL across diverse data sizes.
>
> >ANI dataset, though the level of accuracy and data generation of this dataset is suspect.
>
> We would like to better understand the intention of this critique. The ANI-1x dataset was generated using DFT methods with sufficient accuracy by researchers with extensive expertise, and it has been a widely recognized and extensively used resource in the computational chemistry community. If there are specific concerns or evidence questioning the accuracy or reliability of this dataset, we would greatly appreciate a reference or further elaboration to facilitate a more detailed discussion.
>
> $\textbf{[W2, Q2]: Gradient based force case}$
>
> As mentioned in the “General Remark” to all reviewers, results for models incorporating gradient-based force predictions are provided in Section F.11 and partially summarized in Table 3. These results indicate that PIWSL achieves approximately a 10% reduction in error, even for gradient-based force models.
>
> While we may have misunderstood the reviewer’s intention, we would like to highlight that we also compared PIWSL with a similar loss function that utilizes force and energy labels to estimate perturbed configurations, as described in Equation (A3) and originally proposed by Cooper et al. (2020) [1]. The comparison, presented in Table 5, demonstrates that PIWSL outperforms the approach proposed in [1] for both gradient-based and direct force models.
>
> $\textbf{[W3][W4]: MD simulation analysys}$
>
> As mentioned in the “General Remark,” we would like to clarify that the purpose of Figure 3 is not to assert that the reported model is fully optimized for real-world applications. Instead, it is intended to demonstrate how PIWSL enhances prediction stability during simulations.
>
> >  it is not a good sign that it does not do well for a simple molecule like aspirin.
>
> We hypothesize that the observed stability is more dependent on the model architecture. To further validate the effectiveness of PIWSL, we conducted an additional stability analysis using the PaiNN model with gradient-based force predictions, as shown in the right panel of Figure 3 in the revised manuscript. This analysis demonstrates that simulations can be performed stably for over 100 ps, even when trained on as few as 200 samples, and that PIWSL further improves stability in this scenario.
>
> > Why are the authors doing NVE simulations instead of NVT? Why not try NVT?
>
> In response to this suggestion, we performed a new stability analysis using MD simulations in the canonical statistical ensemble (NVT). The results in Figure A2 show that the thermostat—which maintains a constant temperature—significantly stabilizes the simulations. However, it also obscures the intrinsic stability properties of the MLIP model, which is why we initially chose NVE simulations to assess these characteristics more directly.

---

> ### Author Response · Authors · 2024-11-22
> **reply**
>
> $\textbf{[W5][Q3]: Energy conservation}$
>
> We assume that this comment refers to Appendix F4.2. We intended to convey that NVE statistical sampling is valid when the total energy (along with particle number and volume) remains constant. We acknowledge that the previous description may have been unclear and have revised the corresponding section for clarity.
>
> To illustrate this point, we have added a comparison of the initial and final total energy differences in Figure A1. This comparison shows that total energy conservation is slightly improved, potentially due to the application of PIWSL.
>
> Theoretically, the velocity Verlet algorithm is a symplectic integrator, ensuring the exact conservation of a shadow Hamiltonian as long as the time-step size is less than twice the system's Einstein frequency and the force is derived from the gradient of the potential surface [2]. From this perspective, models utilizing direct force do not meet this criterion, leading to inherently poorer energy conservation during MD simulations compared to gradient-based force models.
>
> $\textbf{[W6]: Inorganic material data set}$
>
> Regarding the claims made in the final line of Section 5.2, we intentionally used the term “indicate” to convey that our assertion is not definitive but rather a suggestion based on the observed results. Additionally, we would like to highlight that we included LMNTO as another example of an inorganic material, ensuring that our conclusions are not solely based on a single dataset.
>
> $\textbf{[W7][Q1] MD22 }$
>
> A. Following the advice, we added new experiments on MD22 using MACE model in Section F1 and Table A10.  Please also refer to “General Remark”.
>
> [1] April M Cooper, Johannes Kästner, Alexander Urban, and Nongnuch Artrith. Efficient training of ann potentials by including atomic forces via taylor expansion and application to water and a transition-metal oxide. npj Comput. Mater., 6:54, 2020
>
> [2] Allen, Michael P., and Dominic J. Tildesley, Computer Simulation of Liquids, 2nd edn (Oxford, 2017; online edn, Oxford Academic, 23 Nov. 2017), https://doi.org/10.1093/oso/9780198803195.001.0001, accessed 21 Nov. 2024.

---

> > ### Comment · Reviewer_kYHe · 2024-11-27
> > **Response to authors**
> >
> > Thank you to the authors for your response and updates. I have read through the updated version of the paper and all of the reviewer responses. In addition to the important points and concerns that Reviewer xuZ7 raised:
> >
> > - Authors mention results from gradient-based force predictions in Section F.11 but I see no Section F.11 in the paper. Is this Section F.10? The authors are measuring the curl of the forces and I think this is not enough to measure energy conservation. A direct measure of energy conservation would be helpful. From the updates in F.4.2, it seems like models both with and without the loss are not conserving energy.
> >
> > - ANI-1x dataset was created using wB97x/6-31G*. This is known among the computational chemistry community to not be very accurate as the basis set is very small. Future work such as AIMNet2 [1] has generated data at a better level of theory to account for this. As reviewer xuZ7 notes, many other datasets are also now available.
> >
> > - I appreciate the additional experiment on ANI. However, PaiNN is  a model that has been far surpassed in performance at this point. How much of this improvement is only because PaiNN is not a very good model? If a better architecture was used, would we still see the same improvement?
> >
> > In conclusion, at its current stage, I do not think this work is ready to be accepted.

---

> > > ### Author Response · Authors · 2024-11-29
> > >
> > > Dear Reviewer kYHe,
> > >
> > > Thank you very much for your insightful and constructive comments. We have carefully considered your feedback and have prepared a thorough response, including additional experimental results to address your points. Notably, we conducted a new experiment using the MACE foundation models (MACE-MP and MACE-OFF) to demonstrate that PIWSL remains effective in the context of foundation models combined with finetuning. Please also check our reply to xuZ7. Below, we provide detailed replies to each of your comments.
> > >
> > > $\textbf{Gradient Force Model:}$
> > >
> > > We apologize for the error in referencing the section—it should be F.10. Regarding energy conservation, as previously explained, the direct force model theoretically does not conserve total energy because the forces do not perfectly emulate the gradient of the potential, which is a limitation inherent to this type of model. Our key point is that PIWSL has the potential to mitigate this issue, as suggested by the trends observed in Figure A1.
> > >
> > > $\textbf{ANI-1x Data set Accuracy:}$
> > >
> > > Thank you for your clarification. We would like to respectfully highlight that the accuracy challenges associated with large datasets are also reported in the SPICE and OpenCatalyst datasets [1,2]. This issue is not unique to the ANI-1x dataset but is instead a broader problem affecting many datasets. Addressing this challenge requires collective effort from the research community. We want to note that improving possible deficiencies of the reference method used to the generate the respective data set is out of this paper's scope. We also do not evaluate the accuracy of the reference level of theory but evaluate how accurately machine-learned potentials can reproduce it. However, we would like to thank the reviewer again for a fruitful information.
> > >
> > > $\textbf{PaiNN:}$
> > >
> > > The PaiNN model used in our experiments, provided by the OpenCatalyst repository, is not a large model. While we have not validated it within the MLIP framework, it is generally observed that larger models benefit more from larger datasets. For example, as shown in our results, the eSCN model trained on TiO2​ with 2000 samples achieves 20–50% improvement, even when the energy RMSE approaches 1 kcal/mol. Unfortunately, validating larger models within this rebuttal is not feasible due to the significant computational resources and training time required.
> > >
> > > To clarify, the key contributions of our paper are as follows:
> > >
> > > 1. Demonstrating performance gains across a wide range of models, datasets, and dataset sizes.
> > > 2. Improving robustness during MD simulations, particularly in data-sparse regimes.
> > > 3. Showcasing the flexibility of PIWSL for extensions and new formulations.
> > > 4. Reducing force prediction errors in scenarios without force labels.
> > >
> > > We hope the reviewer will appreciate not only the first contribution but also the broader significance of the other points. The flexibility of the PIWSL framework opens up avenues for exploring novel formulations, which we believe is a substantial contribution to the field. We also noted that some of the reviewer’s criticism should be attributed to the model capacity and property, not to PIWSL. And we hope that reviewer would kindly take care of the difference when evaluate our method (for example, stability and a poor energy conservation of direct force model during MD simulation, performance improvement beyond model capacity, and the data set size dependence of the PIWSL effectiveness).
> > >
> > > Finally, we respectfully note that the current low evaluation score appears to focus solely on the first contribution, without fully considering our stated assumptions and target scenarios. We believe this does not fairly reflect the overall contributions of the paper.
> > >
> > > Best regards,
> > >
> > > Authors

---

> > > > ### Author Response · Authors · 2024-12-02
> > > >
> > > > Dear Reviewer kYHe,
> > > >
> > > > The discussion period will conclude within the next 24 hours.
> > > > We sincerely appreciate your detailed and constructive feedback. Regarding the concerns mentioned, we believe our most recent responses have addressed these points, at least partially:
> > > >
> > > > 1.   High errors: We provided clarification on the "1 kcal/mol accuracy" and included a new experiment on TiO2 using eSCN.
> > > > 2.   Limited gains at scale: We offered a justification under the "Data Scaling" section.
> > > > 3.    Lack of baselines: We addressed this by referencing the "Competitive Baseline" with Cooper+(2020).
> > > > 4.    Unrealistic limited data setting: We clarified this under "Assumed Scenario in Our Paper" and provided new results with the MACE foundation model with non-negligible error reduction, as a showcase for foundation model + finetuning scenario.
> > > >
> > > > Given these updates, we respectfully feel that the current rating may not fully reflect the paper's contributions, and we are finding it difficult to understand why it would fall significantly below the borderline.
> > > >
> > > > If there are any additional points that remain unaddressed or require further clarification to support a reconsideration of the rating, we would be very grateful if you could share them with us.
> > > >
> > > > Thank you once again for your valuable feedback and guidance.
> > > >
> > > > Best regards,
> > > >
> > > > Authors

---

### Author Response · Authors · 2024-11-22
**General Remark 2**

$\textbf{Further analysis on ML ab-initio MD simulations (NVT and NVE with gradient-based force model): }$

We would like to clarify the purpose of Figure 3. Its intention is not to claim that the trained model is entirely suitable for real-world applications but to demonstrate that PIWSL improves prediction stability during simulations. Indeed, the stability of MLIP models depends strongly on the model architecture. Figure 3 shows models employing a direct force branch with relatively poor stability.

To further validate the effectiveness of PIWSL, we added a new stability analysis using the PaiNN model with gradient-based force predictions. The results are shown in  Figure 3, on the right panel in the revised manuscript. This analysis shows that simulations can be performed stably for up to 100 ps, even when trained on only 200 samples, and that PIWSL further enhances stability in this scenario.

Additionally, in response to the reviewer's feedback, we incorporated a new stability measure using MD simulations in the canonical statistical ensemble (NVT), presented in Figure A2. These results highlight that the thermostat, which is required to maintain a constant temperature, significantly stabilizes simulations and can obscure the inherent stability characteristics of the MLIP model.

Finally, we would like to emphasize that the effectiveness of PIWSL has been thoroughly validated across five diverse datasets (ANI-1x, rMD17, MD22, TiO2, LMNTO) using six different models (SchNet, SpinConv, PaiNN, eSCN, Equiformer v2, MACE), including several state-of-the-art (SOTA) approaches. These validations span a wide range of data sizes, from as few as 50 samples to as many as 5,000,000, and include molecular dynamics (MD) simulations under both NVE and NVT conditions. We hope the reviewers will recognize and appropriately value the scope and rigor of these experiments.

Detailed responses to each review are provided below. We hope that our responses address your question satisfactorily. Please let us know if further clarification is needed.

Best regards,
The Authors

---

### Author Response · Authors · 2024-11-22
**General Remark**

Dear Reviewers,

Thank you very much for your very kind and encouraging suggestions. Based on your valuable suggestions, we uploaded the refined new manuscript, where the newly added or modified text was written in red. Before addressing each review individually, we want to summarize the key points.

$\textbf{The objective of Our Paper:}$

As we emphasized in the abstract, introduction, and conclusion, the primary objective of our work is to introduce a simple yet widely applicable method that reduces errors in the sparse data regime. This is particularly important for real-world computational chemistry applications, where training data for MLIP models should be generated from scratch using DFT solvers. Due to the high computational cost of DFT calculations, minimizing the overall dataset size is crucial. Reducing errors in the sparse data regimes is essential to accelerate this process and enhance sample efficiency. We emphasize that a reduction of the error in sparse data regimes (even though it could be considered as making a "bad" ML model less "bad") is crucial in such scenarios because it minimizes the number of required DFT calculations, increases the length of atomistic simulations with the surrogate (i.e., increases the number of sampled MD frames), and decreases the overall training time for a reliable MLIP model. Thus, this capability to efficiently reduce error represents a significant contribution from a practical point of view. We showed that the PIWSL method can reduce errors across various models, including recent SOTA models, within these sparse data regimes across multiple material datasets.

$\textbf{Training on Larger Data sets (ANI-1x: 1,000,000 and 5,000,000 Samples)}$

In response to reviewers' recommendations, we included a new assessment of the PIWSL method on a large dataset. Specifically, we trained the PaiNN model on 1,000,000 and 5,000,000 samples from the ANI-1x dataset. The results are as follows:

$$\\begin{array} {|r|r|}\\hline  & N_{train} & Farce MAE & Force RMSE & Energy MAE & Energy RMSE \\\ \\hline Baseline & 1,000,000 & 0.67 & 2.74 & 4.90 & 6.56 \\\ \\hline PIWSL &  & 0.68 & 2.77 & \textbf{4.48} & \textbf{6.06} \\\ \\hline Baseline & 5,000,000 & 0.53 & 2.18 & 3.94 & 5.24 \\\ \\hline PIWSL &  & 0.55 & 2.24 & \textbf{3.25} & \textbf{4.75} \\\ \\hline  \\end{array}$$

These results demonstrate that PIWSL remains effective in the large dataset regime (1,000,000 and 5,000,000 samples), reducing the energy error by approximately 10% while maintaining the force prediction error. This table is also included as Table A5 in the revised manuscript.

$\textbf{A New result with MACE model on MD22 data set (buckyball catcher)}$

To address the reviewers' suggestions, we conducted an additional experiment on a large molecule dataset using the state-of-the-art MACE model. For this purpose, we trained the MACE model on the MD22 dataset, focusing on the "buckyball catcher" molecule due to its large size (148 atoms) and complex structure. The results are as follows:

$$\\begin{array} {|r|r|}\\hline  &  & N_{train}=50 &  & N_{train}=600 & \\\ \\hline Model &  & Baseline & PIWSL & Baseline & PIWSL \\\ \\hline MACE & E & 0.163 & \textbf{0.155} & 0.091 & \textbf{0.085} \\\ \\hline  & F & 6.780 & 6.758 & 1.886 & 1.899 \\\ \\hline MACE w.t. SWA & E & 0.150 & \textbf{0.106} & 0.080 & \textbf{0.069} \\\ \\hline  & F & 6.844 & 6.807 & 2.056 & 2.052 \\\ \\hline  \\end{array}$$

These results confirm that PIWSL is effective in the large molecule regime when applied with MACE, particularly in improving energy predictions in the sparse data regime, as expected. This table is also included as Table A10 in the revised manuscript.
(Here we used the set up introduced in the official document [2] with slight modification on max_L and mini-batch size, to accelerate training).

[1] Ilyes Batatia, David Peter Kovacs, Gregor N. C. Simm, Christoph Ortner, and Gabor Csanyi. MACE: Higher Order Equivariant Message Passing Neural Networks for Fast and Accurate Force Fields. Adv. Neural Inf. Process. Syst., 35:11423–11436, 2022.

[2] https://mace-docs.readthedocs.io/en/latest/examples/training_examples.html

---

> ### Comment · Reviewer_xnGz · 2024-11-26
>
> Thank you for your response. I'm glad to see that your proposed method is effective on large datasets. I will consider revising my score.

---

> > ### Author Response · Authors · 2024-11-27
> > **reply**
> >
> > Thank you very much for your kind response!
> > We would like to conduct a further polish of our paper based on the reviewers' feedback.

---

### Author Response · Authors · 2024-11-29
**Additional Experiments**

Dear Reviewers,

Thank you very much for your insightful feedback. To address your comments, we conducted additional experiments to further demonstrate the effectiveness of PIWSL in the following scenarios: (1) fine-tuning foundation models and (2) cases where the MLIP model has already achieved an accuracy of 1 kcal/mol. The detailed results are as follows:

$$\textbf{(1) fine-tuning foundation models}$$

To illustrate that PIWSL remains valuable in the foundation model + fine-tuning framework, we conducted an experiment on the MD22 dataset using MACE as the foundation model, trained with 50 samples. The results are:

$$\\begin{array} {|r|r|}\\hline Model &  & F-MAE & E-MAE \\\ \\hline Baseline & MP & 16.503 \\pm 0.206 & 0.329 \\pm 0.071 \\\ \\hline PIWSL & MP & \\mathbf{16.150 \\pm 0.244} & 0.314 \\pm 0.023 \\\ \\hline Baselie & OFF & 7.981 \\pm 0.090 & 0.181 \\pm 0.024 \\\ \\hline PIWSL & OFF & \\mathbf{7.781 \\pm 0.015} & \\mathbf{0.155 \\pm 0.008} \\\ \\hline  \\end{array}$$

Here, MP refers to MACE-MP ("large"), a universal foundation model, and OFF refers to MACE-OFF ("large\_OFF"), a foundation model specialized for organic materials. We used the setup provided by official repository with increasing total epoch from 10 to 100. The results demonstrate that PIWSL accelerates the learning of force labels, achieving consistently lower errors even under the best-case scenario. Notably, MACE-OFF outperforms MACE-MP, highlighting that MACE-MP struggles to learn from organic material data. This indicates that while foundation models can be effective, their performance is highly dependent on the pretraining dataset, and they are not universally applicable. We believe this observation supports our assumption that scratch training remains valuable for studies involving new materials.

Additionally, we observed that incorporating the second-order term of the Taylor expansion in the PITC loss is critical in this setup. Through simple mathematical derivations,  $\textbf{we extended the PITC loss formulation to approximately account for the second-order term}$, by simply adding $(F(S_{dr}) - F(S))/2 \\cdot dr$ to PITC potential energy estimation. This also demonstrates the flexibility of the PIWSL method based on mathematics and physics.

$$\textbf{(2) 1kcal/mol Accuracy:}$$

We appreciate the reviewer’s suggestion and acknowledge that we may have misunderstood the intention behind their proposal. Regarding accuracy, it is inherently related to the model size, as larger models are better equipped to capture the diversity of larger datasets. In our case, the PaiNN model used was not sufficiently large. This choice was constrained by the rebuttal timeframe, as training larger models, such as Equiformer or eSCN, would have required significantly more time.
To address this point, we performed an additional experiment using eSCN trained on the TiO₂ dataset with 2,000 samples. The results are as follows:

$$\\begin{array} {|r|r|}\\hline  & F-MAE & F-RMSE & E-MAE & E-RMSE \\\ \\hline Baseline & 0.061 \\pm 0.010 & 0.415 \\pm 0.169 & 0.634 \\pm 0.017 & 1.775 \\pm 0.070 \\\ \\hline PIWSL & \\mathbf{0.028 \\pm 0.000} & \\mathbf{0.163 \\pm 0.015} & \\mathbf{0.329 \\pm 0.026} & \\mathbf{0.899 \\pm 0.085} \\\ \\hline  \\end{array}$$
Notably, the energy errors are less than 1 kcal/mol. Moreover, PIWSL achieves an error reduction of approximately 20–50% for both forces and energies.

---

### Meta-Review · Area_Chair_uzTL · 2024-12-20

**Metareview:**

The submission presents a physics-informed weakly supervised method to improve the generalization of machine-learning force field models (MLFFs), especially in situations where labeled data are scarce. Two losses are proposed based on the physics fact that the force is the negative gradient of potential energy. Comprehensive experiments are conducted over datasets and MLFF architectures, and evaluation on realistic use cases of MD simulation is also appreciated. I agree with the reviewers on the appreciation of the contribution to improve MLFF by leveraging physical laws that bridge different quantities and tasks beyond standard supervised training. Nevertheless, there still remain a few insufficiencies that multiple reviewers have raised:
* Since the proposed method does not provide more information than asking the model to be conservative, what are the advantages over learning a gradient-based/conservative force model, and what could be the rationale that the proposed method outperforms conservative force models. I also asked this question myself to the authors. The authors provided multiple pieces of replies, but are mainly empirical demonstrations, and further clarifications on the roles of the losses do not seem to be helpful for answering this question, since a conservative force model is also expected to achieve the effects. I expect more rationales and evidence on why a conservative force model cannot achieve the same effects, hence ruling out the doubt that the improvement comes from other factors.
* Another common challenge is the use case. The major demonstration is the case of scarce labeled data, while multiple reviewers challenged that there are large data in the domain. The authors further provided additional results on large datasets in the rebuttal. Some reviewers (ji8n and xnGz) appreciated the effort and results, hence increased their scores, while Reviewer xuZ7 challenged the significance of the improvements in the large data cases. I agree with the reviewer on this point. There also remains the doubt that the improvement is insignificant when using better model architectures or conservative force models, or finetuning a large foundation model. Nevertheless, while showing the benefit in the large-data case is one way to show the merit, I don't think it is not common that molecular tasks have scarce data, but the authors may need to find a proper realistic setting, e.g., very accurate energy/force labels, hard-to-compute quantities, or costly experimental data.
* Reviewer kYHe also challenged the stability analysis in MD is short and that the MD performance and settings are not satisfying. These two points seem to have been addressed by additional experiments in the rebuttal.

Considering both sides, while the authors have done a hard work that systematically studied the method in various cases, the remaining insufficiencies are still non-negligible. I recommend a reject to the submission for the current form, and encourage the authors to further strengthen the insufficiencies accordingly for a stronger paper.

**Additional Comments On Reviewer Discussion:**

* All reviewers appreciated the contribution to improve MLFF by leveraging physical laws and tasks beyond standard supervised training. I agree with this point.
* Reviewer kYHe asked the question that since the proposed method does not provide more information than asking the model to be conservative, what's the significance of the proposed losses is using a gradient-based/conservative force model. I also asked this question myself to the authors. The authors provided multiple pieces of replies, but are mainly empirical demonstrations, and further clarifications on the roles of the losses do not seem to be helpful for answering this question, since a conservative force model is also expected to achieve the effects. I expect more rationales and evidence on why a conservative force model cannot achieve the same effects, hence ruling out the doubt that the improvement comes from other factors.
* Reviewers ji8n, xnGz, and xuZ7 asked the significance of the method considering that there are large data in the domain. The authors further provided additional results on large datasets in the rebuttal. Some reviewers (ji8n and xnGz) appreciated the effort and results, hence increased their scores, while Reviewer xuZ7 challenged the significance of the improvements in the large data cases. I agree with the reviewer on this point. There also remains the doubt that the improvement is insignificant when using better model architectures or conservative force models, or finetuning a large foundation model. Nevertheless, while showing the benefit in the large-data case is one way to show the merit, I don't think it is not common that molecular tasks have scarce data, but the authors may need to find a proper realistic setting, e.g., very accurate energy/force labels, hard-to-compute quantities, or costly experimental data.
* Reviewer kYHe also challenged the stability analysis in MD is short and that the MD performance and settings are not satisfying. These two points seem to have been addressed by additional experiments in the rebuttal.

---

### Decision · Program_Chairs · 2025-01-22

Reject